# REHEARSAL-FREE CONTINUAL FEDERATED LEARNING WITH SYNERGISTIC REGULARIZATION

## ABSTRACT

Continual Federated Learning (CFL) allows distributed devices to collaboratively learn novel concepts from continuously shifting training data while avoiding *knowledge forgetting* of previously seen tasks. To tackle this challenge, most current CFL approaches rely on extensive rehearsal of previous data. Despite effectiveness, rehearsal comes at a cost to memory, and it may also violate data privacy. Considering these, we seek to apply regularization techniques to CFL by considering their cost-efficient properties that do not require sample caching or rehearsal. Specifically, we first apply traditional regularization techniques to CFL and observe that existing regularization techniques, especially synaptic intelligence, can achieve promising results under homogeneous data distribution but fail when the data is heterogeneous. Based on this observation, we propose a simple yet effective regularization algorithm for CFL named **FedSSI**, which tailors the synaptic intelligence for the CFL with heterogeneous data settings. FedSSI can not only reduce computational overhead without rehearsal but also address the data heterogeneity issue. Extensive experiments show that FedSSI achieves superior performance compared to state-of-the-art methods.

## 1 INTRODUCTION

Federated learning (FL) is to facilitate the collaborative training of a global deep learning model among multiple edge clients while ensuring the privacy of their locally stored data McMahan et al. (2017); Wang et al. (2023a); Liu et al. (2024). Recently, FL has garnered significant interest and found applications in diverse domains, including recommendation systems Yang et al. (2020); Li et al. (2024d) and smart healthcare solutions Xu et al. (2021); Nguyen et al. (2022).

Typically, FL has been studied in a static setting, where the number of training samples does not change over time. However, in realistic FL applications, each client may continuously collect new data and train the local model with streaming tasks, leading to performance degradation on previous tasksYang et al. (2024); Wang et al. (2024). This phenomenon, known as catastrophic forgetting Ganin et al. (2016), poses a significant challenge in the continual learning (CL) paradigm. This challenge is further compounded in FL settings, where *the data on one client remains inaccessible to others, and clients are constrained by memory and computation resources.*

To address this issue, researchers have studied continual federated learning (CFL), which enables each client to continuously learn from local streaming tasks. FedCIL is proposed in Qi et al. (2023) to learn a generative network and reconstruct previous samples for replay, improving the retention of previous information. The authors in Li et al. (2024a) propose to selectively cache important samples from each task and train the local model with both cached samples and the samples from the current task. The authors in Dong et al. (2022; 2023a) focus on the federated class-incremental learning scenario and train a global model by computing additional class-imbalance losses. The study in Ma et al. (2022a) employs supplementary distilled data on both server and client ends, leveraging knowledge distillation to mitigate catastrophic forgetting.

Despite effectiveness, most CFL approaches require each client to cache samples locally, a strategy referred to as data rehearsal. Maintaining the local information on the client side is controversial. On the one hand, clients are often limited by storage resources (typically edge devices) and are unable to cache sufficient previous samples for replay. On the other hand, privacy concerns are non-negligible in the FL setting. For instance, some companies will collect user data to update models in a short

Table 1: Primary Directions of Progress in CFL. Analysis of the recent major techniques in the CFL system with the main contribution. Here, we focus on three common weak points about data rehearsal, computational overhead, and privacy concerns.

| Reference | Contribution | Common Weak Points | | |
|---|---|---|---|---|
| | | Data Rehearsal | Computational Overhead | Privacy Concerns |
| TARGETZhang et al. (2023) | **Synthetic Exemplar** | ✓(Synthetic Sample) | ✓(Distillation, Generator) | ✗ |
| FedCILQi et al. (2023) | **Generation & Alignment** | ✓(Synthetic Sample) | ✓(ACGAN) | ✓(Shared Generator) |
| Re-FedLi et al. (2024a) | **Synergism** | ✓(Cached Sample) | ✗ | ✗ |
| AF-FCLWuerkaixi et al. (2023) | **Accurate Forgetting** | ✓(Synthetic Sample) | ✓(NF Model) | ✓(Shared Generator) |
| SR-FDILLi et al. (2024b) | **Synergism** | ✓(Cached Sample) | ✓(GAN Model) | ✓(Shared Discriminator) |
| GLFCDong et al. (2022) | **Class-Aware Loss** | ✓(Cached Sample) | ✗ | ✓(Proxy Server) |
| FOTBakman et al. (2023) | **Orthogonality** | ✗ | ✓(Multi-Heads,Projection) | ✓(Task-ID) |
| FedWeITYoon et al. (2021) | **Network Extension** | ✗ | ✗ | ✓(Task-ID) |
| CFeDMa et al. (2022b) | **Two-sides KD** | ✓(Distillation Sample) | ✓(Distillation) | ✗ |
| MFCLBabakniya et al. (2024) | **Data-Free KD** | ✓(Synthetic Sample) | ✓(Extra Training) | ✗ |
| FedETLiu et al. (2023) | **Pre-training Backbone** | ✓(Cached Sample) | ✓(Transformer) | ✗ |
| LGADong et al. (2023a) | **Category-Aware** | ✓(Cached Sample) | ✗ | ✓(Proxy Server) |
| Ours | **Regularization** | ✗ | ✗ | ✗ |

period, but this data may contain timestamps and need to be deleted. Meanwhile, generating synthetic samples for replay using generative models may also pose a risk of privacy leakage. As shown in Table 1, we provide a detailed analysis of the latest CFL algorithms and the specific weak points involved. Moreover, existing methods often overlook the issue of data heterogeneity in real-world FL scenarios. Simply applying the same techniques across different clients can lead to decreased model performance when dealing with highly heterogeneous distributed data.

In this paper, we aim to enhance the regularization technique for continual federated learning by adhering to the following three principles: (1) free rehearsal, (2) low computational cost, and (3) robustness to data heterogeneity. In response to the first two principles mentioned above, we observe that a considerable number of traditional regularization techniques in the centralized environment can offer a possible solution intuitively. To verify it, we deploy these techniques into the CFL scenario and conduct extensive experiments, which proves that some of the traditional regularization techniques can indeed achieve fairish results. Despite the advantages of these techniques, we find that directly combining it with CFL fails to maintain stable performance under varying data heterogeneity. For instance, owing to the traditional synaptic intelligence algorithm, each client can prevent catastrophic forgetting in a free rehearsal and low-cost manner by calculating the surrogate loss of different incremental tasks. However, this surrogate loss is merely calculated based on the local data distribution, and in the case of highly heterogeneous data, the local optimization objective of the surrogate loss may not align with that of the global data distribution. It is quite necessary and important to consider and solve the problem of data heterogeneity when deploying algorithms in real-world CFL scenarios.

To address this issue, we propose an enhanced Synaptic Intelligence (SI)-based CFL approach with synergistic regularization named FedSSI. Specifically, in FedSSI, each client will calculate the surrogate loss for previous tasks based not only on information from the local dataset but also on its correlation to the global dataset. We introduce a personalized surrogate model (PSM) for each client, which incorporates global knowledge into local caching so that the surrogate loss reflects both local and global understandings of the data. The SI algorithm is then employed to mitigate catastrophic forgetting, with each client training the local model using both the new task loss and the PSM. Through extensive experiments on various datasets and two types of incremental tasks (Class-IL and Domain-IL), we demonstrate that FedSSI significantly improves model accuracy compared to state-of-the-art approaches. The major contributions of this paper are summarized as follows:

- We provide an in-depth analysis of regularization-based CFL, identifying that existing methods primarily rely on data rehearsal or massive computational overhead, which poses significant challenges in FL settings where resources are constrained and data privacy must be maintained.

- We improve regularization techniques to address catastrophic forgetting and data heterogeneity in CFL. We propose FedSSI, a simple and efficient method that regulates model updates with both local and global information about data heterogeneity.

- We conduct extensive experiments on various datasets and different CFL task scenarios. Experimental results show that our proposed model outperforms state-of-the-art methods by up to 11.52% in terms of final accuracy on different tasks.

## 2 BACKGROUND AND RELATED WORK

**Federated Learning.** Federated Learning (FL) is a technique designed to train a shared global model by aggregating models from multiple clients that are trained on their own local private datasets McMahan et al. (2017); Wang et al. (2023b); Sun et al. (2022). One widely used architecture for FL is FedAvg McMahan et al. (2017), which optimizes the global model by aggregating the parameters of local models trained on private local data. However, traditional FL algorithms like FedAvg face challenges due to data heterogeneity, where the datasets on clients are Non-IID (non-independent and identically distributed), resulting in degraded model performance Jeong et al. (2018b); Liu et al. (2019). To address the Non-IID issue in FL, a proximal term is introduced in the optimization process in Li et al. (2020) to mitigate the effects of heterogeneous and Non-IID data distribution across participating devices. Another approach, federated distillation Jeong et al. (2018a), aims to distill the knowledge from multiple local models into the global model by aggregating only the soft predictions generated by each model. The authors in Lin et al. (2020) proposed a knowledge distillation method that utilizes unlabeled training samples as a proxy dataset. However, these methods primarily address Non-IID static data with spatial heterogeneity, overlooking potential challenges posed by streaming tasks with temporal heterogeneity.

**Continual Learning.** Continual Learning (CL) is a machine learning technique that allows a model to continuously learn from streaming tasks while retaining knowledge gained from previous tasks Hsu et al. (2018); van de Ven & Tolias (2019). This includes task-incremental learning Dantam et al. (2016); Maltoni & Lomonaco (2018), class-incremental learning Rebuffi et al. (2017); Yu et al. (2020), and domain-incremental learning Mirza et al. (2022); Churamani et al. (2021). Existing approaches in CL can be classified into three main categories: replay-based methods Rebuffi et al. (2017); Liu et al. (2020), regularization-based methods Jung et al. (2020); Yin et al. (2020), and parameter isolation methods Long et al. (2015); Fernando et al. (2017). Replay-based methods select representative old samples to retain previously learned knowledge when training on a new task. Regularization-based methods protect existing knowledge from being overwritten by new knowledge by imposing constraints on the loss function of new tasks. Parameter isolation methods typically introduce additional parameters and computations to learn new tasks. Our focus is on the continual federated learning scenario, which combines the principles of federated learning and continual learning.

**Continual Federated Learning.** Continual Federated Learning (CFL) aims to address the learning of streaming tasks in each client by emphasizing the adaptation of the global model to new data while retaining knowledge from past data. Despite its significance, CFL has only recently garnered attention, with Yoon et al. (2021) being a pioneering work in this field. Their research focuses on Task-IL, requiring unique task IDs during inference and utilizing separate task-specific masks to enhance personalized performance. It is studied in Bakman et al. (2023) that projecting the parameters of different tasks onto different orthogonal subspaces prevents new tasks from overwriting previous task parameters. Other studies, such as Ma et al. (2022b), utilize knowledge distillation at the server and client levels using a surrogate dataset. Recently, Li et al. (2024b;a) proposed calculating the importance of samples separately for the local and global distributions, selectively saving important samples for retraining to mitigate catastrophic forgetting in federated incremental learning scenarios. Some research, like Jiang et al. (2021); Dong et al. (2023b), explores CFL in domains beyond image classification. The authors in Li et al. (2024c) adopt the concept of dynamic networks to allow each client to train multiple personalized models based on local resource availability, effectively isolating knowledge between different incremental tasks and merging models for similar knowledge tasks. Most existing works need to cache extra samples with a memory buffer, and methods like Bakman et al. (2023) and Yoon et al. (2021) require the task ID during the inference stage. Our work focuses on regularization-based CFL methods with free rehearsal and low computational cost.

## 3 PROBLEM FORMULATION AND PRELIMINARIES

**Continual Federated Learning.** A typical CFL problem can be formalized by collaboratively training a global model for $K$ total clients with local streaming data. We now consider each client $k$ can only access the local private streaming tasks $\{\mathcal{T}_k^1, \mathcal{T}_k^2, \cdots, \mathcal{T}_k^n\}$. where $\mathcal{T}_k^t$ denotes the $t$-th task of the local dataset. Here $\mathcal{T}_k^t = \sum_{i=1}^{N^t} (x_{k,t}^{(i)}, y_{k,t}^{(i)})$, which has $N^t$ pairs of sample data $x_{k,t}^{(i)} \in X^t$ and corresponding label $y_{k,t}^{(i)} \in Y^t$. We use $X^t$ and $Y^t$ to represent the domain space and label space for the $t$-th task, which has $|Y^t|$ classes and $Y = \bigcup_{t=1}^n Y^t$ where $Y$ denotes the total classes of all time. Similarly, we use $X = \bigcup_{t=1}^n X^t$ to denote the total domain space for tasks of all time.

In this paper, we focus on two types of CL scenarios: (1) Class-Incremental Task: the main challenge lies in when the sequence of learning tasks arrives, the number of the classes may change, i.e., $Y^1 \neq Y^t, \forall t \in [n]$. (2) Domain-Incremental Task: the main challenge lies in when the sequence of tasks arrives, the client needs to learn the new task while their domain shifts, i.e., $X^1 \neq X^t, \forall t \in [n]$. When the $t$-th task comes, the goal is to train a global model $w^t$ overall $t$ tasks $\mathcal{T}^t = \{\sum_{n=1}^t \sum_{k=1}^K \mathcal{T}_k^n\}$, which can be formulated as :

$$w^t = \arg \min_{w \in \mathbb{R}^d} \sum_{n=1}^t \sum_{k=1}^K \sum_{i=1}^{N_k^n} \frac{1}{|\mathcal{T}^t|} \mathcal{L}_{CE} \left( f_w(x_{k,n}^{(i)}), y_{k,n}^{(i)} \right). \tag{1}$$

where $f_w(\cdot)$ is the output of the global model $w$ on the sample and $\mathcal{L}_{CE}(\cdot)$ is the cross-entropy loss.

**Synaptic Intelligence in CFL.** Synaptic Intelligence (SI) is first introduced by Zenke et al. (2017) in 2017 to mitigate catastrophic forgetting when neural networks are trained sequentially on multiple tasks. SI achieves this by estimating the importance of each synaptic weight change during training and penalizing significant changes to weights that are important for previous tasks. This approach allows the model to preserve knowledge from older tasks while still learning new ones. The key idea behind SI is to assign an importance value to each synaptic weight based on its contribution to the overall loss of the model. When training on a new task, the model is penalized for changing weights with high importance for previous tasks, thereby reducing the risk of forgetting.

To deploy the SI algorithm in CFL (FL+SI), each client will calculate the surrogate loss with the aggregated global model during the local training. Assuming that client $k$ receives the $t$-th task, the training-modified loss is given by:

$$\mathcal{L}_{total}(w_k^t) = \mathcal{L}_{new}(w_k^t) + \alpha \mathcal{L}_{sur}(w_k^t) = \mathcal{L}_{new}(w_k^t) + \alpha \sum_i \Omega_{k,i}^t ||w_{k,i}^t - w_i^{t-1}||^2. \tag{2}$$

where $\mathcal{L}_{new}(w_k^t)$ is the original loss function used for training the local model with the new task, and $\mathcal{L}_{sur}(w_k^t)$ denotes the surrogate loss for the previous tasks. $w_{k,i}^t$ represents the $i$-th parameter of local model $w_k^t$ in client $k$ and $w^{t-1}$ represents the optimal weights in the $(t-1)$-th timestamp, estimated based on its importance for previous tasks. Client $k$ utilizes $\Omega_{k,i}^t$ to measure the importance of the $i$-th parameter of the model $w_k^{t-1}$ for the old local data. $\alpha$ is a scaling parameter to trade off previous versus new knowledge. The importance measure $\Omega_{k,i}^t$ is updated after each training iteration based on the sensitivity of the loss function to changes in the $i$-th weight, calculated as:

$$\Omega_{k,i}^t = \sum_{l<t} \frac{s_{k,i}^l}{||w_{k,i}^l - w_{k,i}^{l-1}||^2 + \epsilon}. \tag{3}$$

where $\epsilon > 0$ is a constant to avoid division by zero. The variable $s_{k,i}^l$ measures the contribution of the $i$-th parameter in client $k$ to the change of the loss function $\mathcal{L}_{new}(w_k^t)$. It is calculated as:

$$s_{k,i}^l = \int_{t^{l-1}}^{t^l} \frac{\partial \mathcal{L}_{new}(w_k^l)}{\partial w_{k,i}^l} \cdot \frac{\partial w_{k,i}^l(t)}{\partial t} dt. \tag{4}$$

where $t^{l-1}$ and $t^l$ denote the start and end iteration of the $l$-th task. By summing the absolute values of the gradients over all training iterations, SI estimates the overall importance of each weight for previous tasks. This importance measure is then used to penalize significant changes to those weights during training on new tasks, thus reducing catastrophic forgetting.

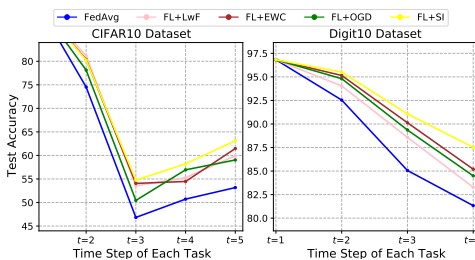 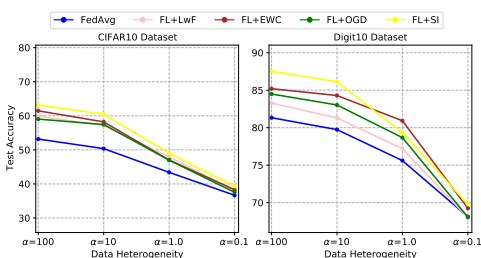

Figure 1: Performance comparisons of various regularization-based CFL methods on CIFAR10 and Digit10 datasets with IID data.

Figure 2: Performance comparisons of aforementioned methods on CIFAR10 and Digit10 datasets with Non-IID data.

## 4 REGULARIZATION TECHNIQUES FOR CONTINUAL FEDERATED LEARNING

Regularization techniques have been known to be cost-efficient for CL without data rehearsal. We in this section analyze several typical regularization techniques used in centralized continual learning and then examine their application in continual federated learning (CFL) with different data heterogeneity settings. Through these examinations, we identify an interesting observation: *existing regularization techniques (especially synaptic intelligence) exhibit great advantages in IID settings but fail to hinder knowledge forgetting in Non-IID settings.* Based on these observations, we then propose a novel method that tailors the synaptic intelligence for the CFL scenario, namely, FedSSI.

### 4.1 DEEP DIVE INTO TRADITIONAL REGULARIZATION TECHNIQUES IN CFL

Based on three principles mentioned in Section 1, here we use four notable algorithms: **LwF** Li & Hoiem (2017), **EWC** Kirkpatrick et al. (2017), **OGD** Farajtabar et al. (2019), and **SI** Zenke et al. (2017) to explore the performance of traditional centralized regularization techniques in the CFL scenario. We simply apply them during the local training process of each client. Each algorithm is combined with the FedAvg McMahan et al. (2017) framework, and we report the final accuracy $A(f)$ after the model completes training on the last streaming task across all tasks. More details about experimental settings can be found in Section 5.1.

**Regularization can mitigate forgetting in decentralized scenarios when data is IID among clients.** As shown in Figure 1, all regularization methods significantly improve test accuracy compared to the baseline, where FL+EWC, FL+OGD, and FL+SI improve the performance by approximately 6%~10%. This suggests that regularization techniques in distributed settings with multiple clients are still effective, indicating their robustness to the restriction of data isolation. Moreover, it is worth noting that FL with SI performs the best among all regularization methods, indicating a great potential to tackle the knowledge-forgetting problem in CFL.

**Regularization fails to mitigate knowledge forgetting when data is Non-IID among clients.** As shown in Figure 2, we observe that as the level of data heterogeneity increases, the performance of all methods significantly declines. Their performance is almost the same as the baseline FedAvg when the data heterogeneity $\alpha = 0.1$, which is a common setting in FL. This phenomenon also occurs in FL+SI, which completely loses its advantage in IID settings. Therefore, to fully unleash the potential of existing regularization methods in CFL, the data heterogeneity should be taken into account when combining them.

### 4.2 FEDSSI: IMPROVED REGULARIZATION TECHNIQUE WITH SYNERGISM FOR CFL

In this paper, we seek to unlock the potential of SI in FL with Non-IID settings considering its advantage in IID scenarios. Specifically, we argue that simply combining SI with FL by calculating the surrogate loss based on the global model will make its performance constrained by that of the global model. When data heterogeneity increases, the performance of the global model significantly decreases, therefore reducing the SI performance. Considering this, we propose that *the surrogate loss in SI should be defined not only based on the importance of the samples in the local dataset but also on their correlation to the global dataset across clients.* This approach ensures that the model can be

---

**Algorithm 1:** FedSSI

---

**Input :** $T$: communication round; $K$: client number; $\eta$: learning rate; $\{\mathcal{T}^t\}_{t=1}^n$: distributed
dataset with $n$ tasks; $w$: parameter of the model; $v_k^t$: personalized surrogate model in
client $k$ for the $t$-th task; $s_{k,i}^l$: contribution of the $i$-th parameter in client $k$ with $t$-th task.

**Output :** $w_1, w_2, \ldots, w_k$: target classification model for each client.

1 **for** $c = 1$ **to** $T$ **do**              // Before the arrival of the $t$-th new task
2      Server randomly selects a subset of devices $S_t$ and send $w^{t-1}$
3      **for** *each selected client* $k \in S_t$ **in parallel do**
4          Update $v_k^{t-1}$ in $s$ local iterations with (5).
5          **for** *During the update of* $v_k^{t-1}$ **do**
6              Calculate the contribution for each parameter $s_{k,i}^l$ with the $(t$-1$)$-th task by (6).
7          **end**
8          Training the target classification model $w_k^t$ with the new task with (2)(1).// Receive
           the $t$-th new task
9          Send the model $w_k^t$ back to the server.
10      **end**
11      $w^t \leftarrow \text{ServerAggregation}(\{w_k^t\}_{k \in S_t})$
12 **end**

---

better trained with the parameter contribution from previously seen tasks. In a standard FL scenario, each client can only access its own local model and the global model, which respectively contain local and global information. A straightforward idea is to calculate two parameter contributions using local and global models and then regularize the local training model based on the summed loss. Building upon this idea, the following capabilities should be added: (1) The global model is aggregated from the local models of participating clients, allowing the surrogate loss of the global model to be calculated locally without training the global model. (2) A control mechanism should be available to adjust the proportion of local and global information. (3) The new module for data heterogeneity will not significantly increase the computational costs or require rehearsal.

Unlike SI algorithms in centralized environments that only consider the data distribution of a single environment, FedSSI introduces a Personalized Surrogate Model (PSM) to balance data heterogeneity across clients. Here, the PSM is not used as the target classification model; it is solely employed to calculate the contribution of parameters. Before clients receive new tasks, a PSM will be trained along with the global model on the current local task. Since this is purely local training assisted by an already converged global model, the training of the PSM is very fast (accounting for only 1/40 of the training cost per task and requiring no communication). We calculate and save the parameter contributions during the local convergence process of the PSM, which can then be locally discarded after its contribution has been computed. Then, each client trains on the new task with the local model and parameter contribution scores. Suppose that the client $k$ receives the global model $w^{t-1}$ before the arrival of the $t$-th new task, and the clients update PSM $v_k^{t-1}$ with the current local samples $\mathcal{T}_k^{t-1}$ in $s$ iterations as follows:

$$v_{k,s}^{t-1} = v_{k,s-1}^{t-1} - \eta\left(\sum_{i=1}^M \nabla\mathcal{L}\left(f_{v_{k,s-1}^{t-1}}(x_{k,t-1}^{(i)}), y_{k,t-1}^{(i)}\right) + q(\lambda)(v_{k,s-1}^{t-1} - w^{t-1})\right). \quad (5)$$

where $q(\lambda) = \frac{1-\lambda}{2\lambda}, \lambda \in (0,1)$, and $\eta$ is the rate to control the step size of the update. The hyper-parameter $\lambda$ adjusts the balance between the local and global information in the update.

To better understand the update process, we can draw an analogy to momentum methods in optimization. Momentum-based methods leverage past updates to guide the current update direction Chan et al. (1996). Similarly, the term $q(\lambda)(v_{k,s-1}^{t-1} - w^{t-1})$ acts as a momentum component. It incorporates information from the global model $w^{t-1}$ to influence the personalized surrogate model (PSM) $v_k^{t-1}$. The hyper-parameter $\lambda \in (0,1)$ controls the weight of this momentum component. Denote that $\alpha$ refers to the degree of data heterogeneity and when $\alpha$ has a higher value, indicating a trend towards homogeneity in distribution, clients need to focus more on local knowledge. This means that by setting a larger $\lambda$ value, PSM can rely more on local knowledge. Conversely, as $\lambda$ decreases, the

emphasis shifts more towards learning global knowledge with heterogeneous data distribution. Based on this, we can theoretically judge there exists a positive correlation between $\alpha$ and $\lambda$, which means that the data heterogeneity will be addressed by controlling the $\lambda$ value.

Then, each client $k$ leverages the PSM $v_k^{t-1}$ to compute the parameter contribution:

$$s_{k,i}^l = \int_{t^{l-1}}^{t^l} \frac{\partial \mathcal{L}_{new}(v_k^l)}{\partial v_{k,i}^l} \cdot \frac{\partial v_{k,i}^l(t)}{\partial t} dt. \tag{6}$$

Considering $v_k$ accommodates the local and global knowledge simultaneously, this refined contribution $s_k^l$ is expected to achieve a better balance of memorizing knowledge in both the previous global model and in the new data. Next, client $k$ uses (3) to compute $\Omega_k^t$ for each parameter $i$ with $s_k^l, \forall l=1,\ldots,t-1$. Finally, client $k$ establishes the local loss with (2) and trains the local model $w_k^t$. The algorithm of FedSSI can be found in Alg.1.

**Discussion about unique challenges.** The unique challenges in this paper are more comprehensive yet practical than existing CFL works. In CFL, there are only a handful of works that successfully tackle both catastrophic forgetting and data heterogeneity simultaneously. In addition, these existing works often rely on substantial resource expenditures and risk privacy breaches, employing techniques such as generative replay, memory buffers, and dynamic networks, primarily traditional centralized methods that overlook the resource constraints of client devices.

In this paper, we have explored the contradiction between the resource constraints of clients and the generally higher resource costs in CL through empirical experiments. Our paper takes a different approach, starting by considering the resource limitations of clients in FL and leveraging appropriate technologies. From this foundation, we aim to solve both catastrophic forgetting and data heterogeneity challenges concurrently.

### 4.3 Analytical Understanding of the Personalized Surrogate Model in FedSSI

In this section, we provide both the effectiveness and the convergence of personalized surrogate models. To simplify the notation, here we conduct an analysis on a fixed task while the convergence does not depend on the CL setting.

**Definition 1** (Personalized Surrogate Model Formulation.) Denote the objective of personalized informative model $v_k$ on client $k$ while $f(\cdot)$ is strongly convex as:

$$\hat{v}_k(\lambda) := \arg\min_{v_k} \left\{ f(v_k) + \frac{q(\lambda)}{2} ||v_k - \hat{w}||^2 \right\}, \quad \text{where } q(\lambda) := \frac{1-\lambda}{2\lambda}, \ \lambda \in (0,1). \tag{7}$$

where $\hat{w}$ denotes the global model.

**Proposition 1** (Proportion of Global and Local Information.) *For all $\lambda \in (0,1)$, $\lambda \to f(\lambda)$ is non-increasing:*

$$\frac{\partial f(\hat{v}_k(\lambda))}{\partial \lambda} \leq 0, \qquad \frac{\partial ||\hat{v}_k(\lambda) - \hat{w}||^2}{\partial \lambda} \geq 0. \tag{8}$$

Then, for $k \in [K]$, we can get:

$$\lim_{\lambda \to 0} \hat{v}_k(\lambda) := \hat{w}. \tag{9}$$

*Proof.* The proof here directly follows the proof in Theorem 3.1 Hanzely & Richtárik (2020). As $\lambda$ declines and $q(\lambda)$ grows, the objective of Eq. 7 tends to optimize $||v_k - \hat{w}||^2$ and increase the local empirical training loss $f(v_k)$, leading to the convergence on the global model. Hence, we can modify the $\lambda$ value to adjust the optimization direction of our model $v_k$, thus the dominance of local and global model information.

**Theorem 1** (Convergence of Personalized Surrogate ModelLi et al. (2024a).) *Assuming $w^t$ converges to the optimal model $\hat{w}$ with convergence rate $g(t)$ for each client $k \in [K]$, such that $\mathbb{E}\left[||w^t - \hat{w}||^2\right] \leq g(t)$ and $\lim_{t\to\infty} g(t) = 0$. There exists a constant $C < \infty$ ensuring that the personalized surrogate model $v_k^t$ converges to its optimal counterpart $\hat{v}_k$ at a rate proportional to $Cg(t)$.*

Table 2: Performance comparison of various methods in two incremental scenarios.

| Method | CIFAR10 | | CIFAI100 | | Tiny-ImageNet | | Digit10 | | Office31 | | Office-Caltech-10 | |
|---|---|---|---|---|---|---|---|---|---|---|---|---|
| | $A(f)$ | $\bar{A}$ | $A(f)$ | $\bar{A}$ | $A(f)$ | $\bar{A}$ | $A(f)$ | $\bar{A}$ | $A(f)$ | $\bar{A}$ | $A(f)$ | $\bar{A}$ |
| FedAvg | 36.68±1.32 | 59.17±0.08 | 27.15±0.87 | 41.36±0.24 | 30.16±0.19 | 50.65±0.11 | 68.12±0.04 | 80.34±0.02 | 48.97±0.74 | 56.29±1.15 | 55.41±0.52 | 57.61±0.93 |
| FedProx | 35.88±0.92 | 59.20±0.13 | 27.84±0.65 | 40.92±0.14 | 29.04±0.53 | 49.93±0.75 | 68.95±0.08 | 80.26±0.09 | 46.33±0.16 | 54.03±0.77 | 53.90±0.43 | 56.10±0.44 |
| FL+LwF | 38.04±0.33 | 59.93±0.25 | 31.91±0.40 | 42.56±0.57 | 34.58±0.29 | 52.91±0.19 | 67.99±0.12 | 80.02±0.06 | 50.70±0.19 | 57.20±0.63 | 57.11±0.28 | 59.75±0.74 |
| FL+EWC | 38.31±0.02 | 60.19±0.10 | 33.36±0.79 | 43.25±0.27 | 36.15±0.11 | 53.87±0.04 | 69.25±0.26 | 81.54±0.15 | 52.24±0.61 | 57.91±0.35 | 58.69±0.50 | 60.06±0.75 |
| FL+OGD | 37.55±0.78 | 59.88±0.42 | 32.87±0.39 | 43.56±0.36 | 35.71±0.42 | 53.19±0.21 | 68.07±0.33 | 79.95±0.05 | 51.86±0.37 | 58.10±0.61 | 58.01±0.81 | 60.20±0.53 |
| FL+SI | 39.32±1.01 | 60.96±0.37 | 33.72±0.78 | 43.82±0.29 | 35.87±0.51 | 53.65±0.34 | 69.79±0.56 | 80.92±0.08 | 53.10±1.11 | 58.28±0.27 | 51.82±0.94 | 60.27±0.48 |
| Re-Fed | 38.08±0.46 | 59.02±0.31 | 32.95±0.31 | 42.50±0.18 | 33.43±0.54 | 51.98±0.32 | 67.85±0.37 | 79.85±0.25 | 50.11±0.29 | 57.46±0.34 | 59.16±0.40 | 60.01±0.33 |
| FedCIL | 37.96±1.68 | 58.30±1.22 | 30.88±1.04 | 42.16±0.97 | 31.35±1.27 | 50.93±0.84 | 68.17±0.85 | 80.02±0.63 | 49.15±0.92 | 56.78±0.79 | 57.80±0.74 | 59.13±0.52 |
| GLFC | 38.43±1.43 | 60.03±1.16 | 33.17±0.62 | 43.28±0.81 | 32.11±0.40 | 51.79±0.54 | 67.39±0.88 | 78.53±0.47 | 48.30±0.53 | 55.82±0.38 | 58.24±0.65 | 59.77±0.40 |
| FOT | 40.18±1.26 | 61.41±0.66 | 36.15±0.40 | 43.14±0.51 | 37.23±0.14 | 54.87±0.22 | 68.54±0.38 | 79.70±0.11 | 49.12±0.83 | 56.17±0.44 | 60.30±0.13 | 60.76±0.98 |
| FedWeIT | 37.96±0.52 | 59.89±0.12 | 35.84±1.05 | 44.20±0.45 | 34.98±0.93 | 53.04±0.71 | 69.71±0.20 | 80.91±0.09 | 51.49±0.64 | 57.83±0.89 | 58.53±0.49 | 59.72±0.70 |
| FedSSI | **42.58±0.79** | **62.65±0.13** | **37.96±0.35** | **45.28±0.18** | **40.56±0.58** | **56.80±0.20** | **72.09±0.41** | **82.49±0.03** | **55.28±0.98** | **60.05±0.45** | **62.57±0.86** | **62.94±0.36** |

Here we introduce Theorem 1 proposed by Li et al. (2024a), which is stated by our proposed personalized surrogate model. Based on it, we ensure the convergence of the PSM and the effectiveness of FedSSI can be proved along by Proposition 1.

# 5 EXPERIMENTS

## 5.1 EXPERIMENT SETUP

**Datasets.** We conduct our experiments with heterogeneously partitioned datasets across two federated incremental learning scenarios using six datasets: (1) **Class-Incremental Learning:** CIFAR10 Krizhevsky et al. (2009), CIFAR100 Krizhevsky et al. (2009), and Tiny-ImageNet Le & Yang (2015); (2) **Domain-Incremental Learning:** Digit10, Office31 Saenko et al. (2010), and Office-Caltech-10 Zhang & Davison (2020). The Digit10 dataset contains 10 digit categories in four domains: MNIST LeCun et al. (2010), EMNIST Cohen et al. (2017), USPS Hull (1994), and SVHN Netzer et al. (2011). Details of the datasets and data processing can be found in Appendix A.

**Baseline.** For a fair comparison with other key works, we follow the same protocols proposed by McMahan et al. (2017); Rebuffi et al. (2017) to set up FIL tasks. We evaluate all methods using two representative FL models: **FedAvg** McMahan et al. (2017) and **FedProx** Li et al. (2020); two models designed for continual federated learning without data rehearsal: **FOT** Bakman et al. (2023) and **FedWeIT** Yoon et al. (2021); three models designed for continual federated learning with data rehearsal: **Re-Fed** Li et al. (2024a), **FedCIL** Qi et al. (2023), and **GLFC** Dong et al. (2022); and four custom methods combining traditional CL techniques with the FedAvg algorithm: **FL+LwF** Li & Hoiem (2017), **FL+EWC** Kirkpatrick et al. (2017), **FL+OGD** Farajtabar et al. (2019), and **FL+SI** Zenke et al. (2017). Details of the baselines and data processing can be found in Appendix B.

**Configurations.** Unless otherwise mentioned, we employ ResNet18 He et al. (2016) as the backbone model in all methods and use the Dirichlet distribution $\text{Dir}(\alpha)$ to distribute local samples, inducing data heterogeneity for all tasks, where a smaller $\alpha$ indicates higher data heterogeneity. For a fair comparison, we set the memory buffer 300 for each client in models designed for continual federated learning with data rehearsal to cache synthetic or previous samples. We report the final accuracy $A(f)$ upon completion of the last streaming task and the average accuracy $\bar{A}$ across all tasks. Each experiment set is run twice, and we take each run's final 10 rounds' accuracy to calculate the average value and standard variance. We use Adam as the optimizer with a linear learning rate schedule. All experiments are run on 8 RTX 4090 GPUs and 16 RTX 3090 GPUs. Detailed settings and benchmark parameters are illustrated in Table 6. (In Appendix C)

## 5.2 PERFORMANCE OVERVIEW

**Test Accuracy.** Table 2 shows the test accuracy of various methods with data heterogeneity across six datasets. We report both the final accuracy and average accuracy of the global model when all clients finish their training on all tasks. FOT outperforms other baselines on three class-IL datasets as the orthogonal projection in FOT exhibits effective training with obvious boundaries between streaming

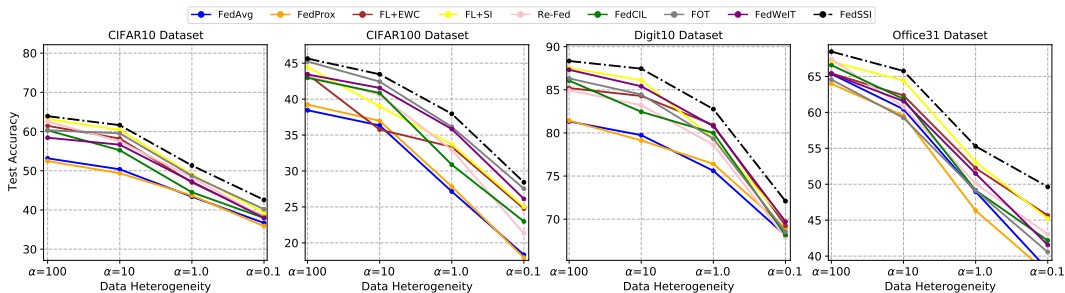

Figure 3: Performance w.r.t data heterogeneity $\alpha$ for four datasets.

Table 3: Test accuracy for FedSSI w.r.t data heterogeneity $\alpha$ and hyper-parameter $\lambda$ on CIFAR10, CIFAR100 and Office31.

| Dataset | $\alpha = 0.1$ | | | $\alpha = 1.0$ | | | $\alpha = 10.0$ | | | $\alpha = 100$ | | |
|---|---|---|---|---|---|---|---|---|---|---|---|---|
| | $\lambda = 0.2$ | $\lambda = 0.5$ | $\lambda = 0.8$ | $\lambda = 0.2$ | $\lambda = 0.5$ | $\lambda = 0.8$ | $\lambda = 0.2$ | $\lambda = 0.5$ | $\lambda = 0.8$ | $\lambda = 0.2$ | $\lambda = 0.5$ | $\lambda = 0.8$ |
| CIFAR10 | **42.58**±0.79 | 41.22±1.03 | 39.93±0.88 | **51.34**±0.88 | 51.19±0.77 | 49.23±0.55 | 60.05±0.88 | 61.01±0.41 | **61.63**±0.46 | 61.28±0.43 | 63.04±0.19 | **63.92**±0.25 |
| CIFAR100 | **28.43**±1.08 | 27.39±1.27 | 26.41±1.20 | 35.81±0.44 | **37.96**±0.35 | 36.01±0.36 | 41.28±0.51 | 42.67±0.99 | **43.45**±1.12 | 44.79±0.59 | 45.23±0.38 | **45.67**±0.46 |
| Office31 | **49.64**±0.33 | 48.56±0.32 | 46.96±0.40 | 54.60±0.48 | **55.28**±0.98 | 54.37±0.62 | 64.88±0.47 | 65.19±0.60 | **65.76**±0.73 | 67.03±0.36 | 67.79±0.24 | **68.45**±0.27 |

tasks. However, its performance experiences a significant decline with domain-IL datasets, where the classes of each task remain unchanged. All regularization-based methods work and CFL models with data rehearsal fail to achieve ideal results with limited memory buffer. FedSSI achieves the best performance in all cases by up to 11.52% in terms of final accuracy. More discussions and results on model performance and communication efficiency are available in Appendix E.

**Data Heterogeneity.** Figure 3 displays the test accuracy with different levels of data heterogeneity on four datasets. As shown in this figure, all methods achieve an improvement in test accuracy with the decline in data heterogeneity, and FedSSI consistently achieves a leading and stable improvement in performance with different levels of data heterogeneity.

**Resource Consumption.** Table 5 lists the training time of the four methods mentioned above. Although most traditional regularization algorithms are not suitable for resource-constrained

Table 5: Training resource overhead for aforementioned methods on CIFAR10. Here, we indicate the time cost and provide the main cause for the additional overhead.

| Method | Time Cost | Main Cause |
|---|---|---|
| FedAvg | 3.13h | - |
| FL+LwF | 4.46h | Knowledge Distillation |
| FL+EWC | 5.75h | Fish Information Matrix |
| FL+OGD | 7.09h | Orthogonal Projection |
| FedSSI | 3.75h | Surrogate Loss |

edge devices in the CFL scenario due to their computational efficiency overhead, we also observe that a few methods that could potentially be deployed in CFL, such as SI, have a relatively low time overhead, achieving significant performance improvements with less than 20% increase in training time compared to fine-tuning (FedAvg).

Table 4 compares the communication efficiency of various methods by measuring the trade-offs between communication rounds and accuracy. Compared to other CFL methods, FedSSI may introduce additional communication rounds but can achieve better performance with a cost-effective trade with a large $\Delta$ value. We observed that FedSSI can achieve a significant $\Delta$ value for all datasets except Digit10, indicating that our communication method is efficient and can bring considerable performance improvements. For Digit10, a possible reason is that the dataset itself has simple features (one-channel images), and all methods can achieve relatively good accuracy. In such cases, the percentage increase in accuracy by FedSSI is relatively small. However, achieving a high-performance improvement on a simple dataset is not easy in itself. In the next version, we will consider using a smaller network model (as learning Digit10 with a CNN is sufficient) for verification. The communication rounds for each task of different datasets are listed in Table 5.1.

**Hyper-parameter.** Then, we conduct more research on the setting of hyper-parameter $\lambda$. In our framework, we modify the $\lambda$ value to adjust the global and local information proportion in PSM. As

Table 4: Evaluation of various methods in terms of the communication rounds to reach the best test accuracy. We report the sum of communication rounds required to achieve the best performance on each task and evaluate with the trade-offs between communication rounds and accuracy. We denote "$\Delta$" as the difference between the accuracy improvement percentage and the round increase percentage of FedSSI and other baselines.

| Method | CIFAR10 | | CIFAR100 | | Tiny-ImageNet | | Digit10 | | Office31 | | Office-Caltech-10 | |
|---|---|---|---|---|---|---|---|---|---|---|---|---|
| | Rounds | $\Delta$ | Rounds | $\Delta$ | Rounds | $\Delta$ | Rounds | $\Delta$ | Rounds | $\Delta$ | Rounds | $\Delta$ |
| FedAvg | 304±2.31 | 16.09%↑ | 839±1.67 | 44.70%↑ | 893±2.93 | 33.03%↑ | 208±1.58 | 2.35%↓ | 165±0.94 | 7.43%↑ | 132±1.25 | 4.59%↑ |
| FedProx | 315±1.84 | 22.17%↑ | 852±1.76 | 42.69%↑ | 900±2.28 | 39.00%↑ | 204±1.51 | 5.74%↓ | 166±1.06 | 14.50%↑ | 145±0.99 | 17.46%↑ |
| FL+LwF | 300±1.32 | 10.60%↑ | 832±2.17 | 23.05%↑ | 897±1.41 | 16.29%↑ | 211±1.04 | 0.60%↓ | 162±0.98 | 1.63%↑ | 134±0.71 | 2.84%↑ |
| FL+EWC | 324±1.12 | 17.32%↑ | 810±2.25 | 15.27%↑ | 909±2.16 | 12.53%↑ | 232±1.81 | 7.12%↑ | 166±0.66 | 1.00%↑ | 150±1.32 | 11.28%↑ |
| FL+OGD | 309±1.65 | 15.01%↑ | 833±2.34 | 19.69%↑ | 900±2.10 | 12.91%↑ | 213±1.19 | 0.27%↑ | 164±1.52 | 0.50%↑ | 148±0.69 | 11.24%↑ |
| FL+SI | 296±1.11 | 5.59%↑ | 816±1.97 | 14.78%↑ | 897±2.20 | 12.07%↑ | 212±0.87 | 2.84%↓ | 165±1.32 | 1.35%↓ | 159±0.85 | 30.81%↑ |
| Re-Fed | 319±1.23 | 16.52%↑ | 844±2.15 | 20.66%↑ | 894±1.92 | 19.99%↑ | 222±1.48 | 4.90%↑ | 168±0.98 | 6.75%↑ | 145±0.76 | 7.14%↑ |
| FedCIL | 323±2.78 | 18.05%↑ | 866±2.25 | 30.78%↑ | 903±1.94 | 29.05%↑ | 220±1.42 | 3.48%↑ | 168±1.07 | 8.90%↑ | 147±0.91 | 10.97%↑ |
| GLFC | 312±1.47 | 13.36%↑ | 830±2.03 | 18.30%↑ | 901±2.12 | 25.76%↑ | 220±0.96 | 4.70%↑ | 166±1.14 | 9.63%↑ | 158±1.31 | 16.93%↑ |
| FOT | 306±2.26 | 6.63%↑ | 840±1.35 | 10.01%↑ | 903±1.09 | 8.61%↑ | 220±0.76 | 2.91%↑ | 170±0.98 | 10.19%↑ | 158±1.04 | 13.26%↑ |
| FedWeIT | 312±1.95 | 14.73%↑ | 870±1.58 | 14.19%↑ | 880±2.26 | 13.00%↑ | 217±1.31 | 0.27%↓ | 165±1.79 | 1.91%↑ | 141±1.52 | 5.48%↑ |
| FedSSI | 304±1.29 | / | 798±2.03 | / | 906±1.62 | / | 225±1.34 | / | 174±0.80 | / | 143±0.81 | / |

shown in Table 3, we select three different $\lambda$ values with four different data heterogeneity settings and evaluate the final test accuracy on three datasets. Experimental results show that the value of $\lambda$ should be chosen accordingly under different data heterogeneity. Nevertheless, results exhibit the same trend: as the degree of data heterogeneity increases, FedSSI performs better while $\lambda$ decreases as PSM contains more global information. Empirically, striking a balance between global and local information is the key to addressing the data heterogeneity in CFL. Vice versa, as $\alpha$ increases, the data distribution on the client side becomes more IID. At this point, the clients require less global information and can rely more on their local information for caching important samples. Although we cannot directly relate $\alpha$ and $\lambda$ with a simple formula due to the complexity of the problem, even in specialized research on personalized federated learning (PFL), methods such as Gaussian mixture modeling are relied upon. we can empirically and theoretically judge there exists a positive correlation between $\alpha$ and $\lambda$ with Proposition 1 and Table 3.

**Quantitative Analysis**. Figure 4 shows the qualitative analysis of the number of incremental tasks $n$ on three class-incremental datasets. According to these curves, we can easily observe that our model performs better than other baselines across all tasks, with varying numbers of incremental tasks. It demonstrates that FedSSI enables clients to learn new incremental classes better than other methods.

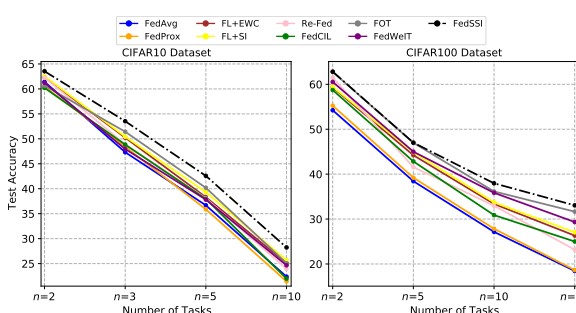

Figure 4: Performance w.r.t number of incremental tasks $n$ for two class-incremental datasets

# 6 CONCLUSION AND FUTURE WORK

In this paper, we introduced FedSSI, a novel regularization-based method for continual federated learning (CFL) that mitigates catastrophic forgetting and manages data heterogeneity without relying on data rehearsal or excessive computational resources. Our extensive experiments show that FedSSI outperforms existing CFL approaches, achieving superior accuracy and stability in resource-constrained environments. This work paves the way for more practical and efficient CFL deployments in real-world scenarios.

Although we have invested in the effectiveness of regularization-based methods over the CFL scenarios without relying on data rehearsal or excessive computational resources, the overhead of other training resources should be taken into account. To deploy the FL system in practical settings, it is necessary to consider resource factors such as training efficiency, model capacity, and even sparsely labeled data. In the future, we seek to work a step forward in this field.

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

APPENDIX

## A DATASETS

**Class-Incremental Task Dataset:** New classes are incrementally introduced over time. The dataset starts with a subset of classes, and new classes are added in subsequent stages, allowing models to learn and adapt to an increasing number of classes.

*(1) CIFAR10:* A dataset with 10 object classes, including various common objects, animals, and vehicles. It consists of 50,000 training images and 10,000 test images.

*(2) CIFAR100:* Similar to CIFAR10, but with 100 fine-grained object classes. It has 50,000 training images and 10,000 test images.

*(3) Tiny-ImageNet:* A subset of the ImageNet dataset with 200 object classes. It contains 100,000 training images, 10,000 validation images, and 10,000 test images.

**Domain-Incremental Task Dataset:** New domains are introduced gradually. The dataset initially contains samples from a specific domain, and new domains are introduced at later stages, enabling models to adapt and generalize to new unseen domains.

*(1) Digit10:* Digit-10 dataset contains 10 digit categories in four domains: MNISTLeCun et al. (2010), EMNISTCohen et al. (2017), USPSHull (1994), SVHNNetzer et al. (2011).Each dataset is a digit image classification dataset of 10 classes in a specific domain, such as handwriting style.

- MNIST: A dataset of handwritten digits with a training set of 60,000 examples and a test set of 10,000 examples.
- EMNIST: An extended version of MNIST that includes handwritten characters (letters and digits) with a training set of 240,000 examples and a test set of 40,000 examples.
- USPS: The United States Postal Service dataset consists of handwritten digits with a training set of 7,291 examples and a test set of 2,007 examples.
- SVHN: The Street View House Numbers dataset contains images of house numbers captured from Google Street View, with a training set of 73,257 examples and a test set of 26,032 examples.

*(2) Office31:* A dataset with images from three different domains: Amazon, Webcam, and DSLR. It consists of 31 object categories, with each domain having around 4,100 images.

*(3) Office-Caltech-10:* A dataset with images from four different domains: Amazon, Caltech, Webcam, and DSLR. It consists of 10 object categories, with each domain having around 2,500 images.

## B BASELINES

**Representative FL methods in CFL:**

- **FedAvg:** It is a representative federated learning model that aggregates client parameters in each communication. It is a simple yet effective model for federated learning.
- **FedProx:** It is also a representative federated learning model, which is better at tackling heterogeneity in federated networks than FedAvg.

**Traditional Regularization techniques in CFL:**

- **FL+LwF:** This method integrates Federated Learning with Learning without Forgetting (LwF). LwF helps mitigate catastrophic forgetting by retaining knowledge from previous tasks while learning new ones. In our implementation, we use FedAvg to aggregate parameters and incorporate LwF to preserve previous knowledge during training.
- **FL+EWC:** This method integrates Federated Learning with Elastic Weight Consolidation (EWC). EWC addresses catastrophic forgetting by imposing a quadratic penalty on the changes to parameters that are important for previously learned tasks, thereby preserving the

knowledge acquired from previous tasks while enabling the model to learn new information efficiently.

- **FL+OGD:** This method combines Federated Learning with Orthogonal Gradient Descent (OGD). OGD mitigates catastrophic forgetting by projecting the gradient updates orthogonally to the subspace spanned by the gradients of previous tasks, thus preserving knowledge of previously learned tasks while allowing new information to be integrated effectively.

- **FL+SI:** This method integrates Federated Learning with Synaptic Intelligence (SI). SI computes the importance of each parameter in a similar manner to EWC but uses a different mechanism for accumulating importance measures over time. We adapt the FedAvg algorithm to include SI, maintaining a balance between learning new tasks and retaining old knowledge.

**Methods designed for CFL:**

- **Re-Fed:** This method deploys a personalized, informative model at each client to assign data with importance scores. It is prior to cache samples with higher importance scores for replay to alleviate catastrophic forgetting.

- **FedCIL:** This approach employs the ACGAN backbone to generate synthetic samples to consolidate the global model and align sample features in the output layer with knowledge distillation.

- **GLFC:** This approach addresses the federated class-incremental learning and trains a global model by computing additional class-imbalance losses. A proxy server is introduced to reconstruct samples to help clients select the best old models.

- **FOT:** This method uses orthogonal projection technology to project different parameters into different spaces to achieve isolation of task parameters. In addition, this method proposes a secure parameter aggregation method based on projection. In the model inference stage, the method requires assuming that the boundaries of the task are known. We modify it to automated inference of the model.

- **FedWeIT:** This method divides parameters into task-specific parameters and shared parameters. A multi-head model based on known task boundaries is used in the original method. To ensure a fair comparison, we modify it to automate the inference of the model.

## C  CONFIGURATIONS

Table 6: Experimental Details. Settings of different datasets in the experiments section.

| Attributes | CIFAR10 | CIFAR100 | Tiny-ImageNet | Digit10 | Office31 | Office-Caltech-10 |
|---|---|---|---|---|---|---|
| Task size | 178MB | 178MB | 435MB | 480M | 88M | 58M |
| Image number | 60K | 60K | 120K | 110K | 4.6k | 2.5k |
| Image Size | $3\times32\times32$ | $3\times32\times32$ | $3\times64\times64$ | $1\times28\times28$ | $3\times300\times300$ | $3\times300\times300$ |
| Task number | $n=5$ | $n=10$ | $n=10$ | $n=4$ | $n=3$ | $n=4$ |
| Task Scenario | Class-IL | Class-IL | Class-IL | Domain-IL | Domain-IL | Domain-IL |
| Batch Size | $s=64$ | $s=64$ | $s=128$ | $s=64$ | $s=32$ | $s=32$ |
| ACC metrics | Top-1 | Top-1 | Top-10 | Top-1 | Top-1 | Top-1 |
| Learning Rate | $l=0.01$ | $l=0.01$ | $l=0.001$ | $l=0.001$ | $l=0.01$ | $l=0.01$ |
| Data heterogeneity | $\alpha=0.1$ | $\alpha=1.0$ | $\alpha=10.0$ | $\alpha=0.1$ | $\alpha=1.0$ | $\alpha=1.0$ |
| Client numbers | $C=20$ | $C=20$ | $C=20$ | $C=15$ | $C=10$ | $C=8$ |
| Local training epoch | $E=20$ | $E=20$ | $E=20$ | $E=20$ | $E=20$ | $E=15$ |
| Client selection ratio | $k=0.4$ | $k=0.5$ | $k=0.6$ | $k=0.4$ | $k=0.4$ | $k=0.5$ |
| Communication Round | $T=80$ | $T=100$ | $T=100$ | $T=60$ | $T=60$ | $T=40$ |

## D  LIMITATION

The limitation of FedSSI is that the PSM may introduce additional storage. However, the model for an FL task is practically not large as edge clients are mostly resource-limited, and the memory

demand by PSM is relatively similar to existing methods like FedProx as to manipulate over an extra model that has the same size as the federated model. In addition, we propose a possible strategy to address this issue: If an FL task has large models and limited memory left on the edge, a prerequisite can be reasonably assumed satisfied: transmission capacity (probably after optimization) is sufficient for the system. In such a case, we can record the PSM model in the server, which is only downloaded and updated locally to compute parameter contributions and is then uploaded to the server again.

# E  ADDITIONAL RESULTS

In this section, we first provide the experiments about scalability and bandwidth constraints to validate the effectiveness of our method. Then, we provide more details about the experiment results of each task. We record the test accuracy of the global model at the training stage of each task and the communication rounds required to achieve the corresponding performance.

## E.1  DETAILED RESULTS OF SCALABILITY AND BANDWIDTH CONSTRAINTS

Table 7 shows the results of test accuracy on scalability and bandwidth constraints. We highlight the **best** result in bold. To verify scalability, we conducted experiments with 100 clients and a client selection rate of 10% ($\alpha = 10.0$). To investigate the impact of bandwidth on our method, we reduced the client selection rate by half and decreased the total number of communication rounds.

Table 7: Performance comparison of various methods with scalability and bandwidth constraints.

| Method | Scalability | | | | Bandwidth | | | |
| --- | --- | --- | --- | --- | --- | --- | --- | --- |
| | CIFAR10 | | Digit10 | | CIFAR10 | | Digit10 | |
| | $A(f)$ | $\bar{A}$ | $A(f)$ | $\bar{A}$ | $A(f)$ | $\bar{A}$ | $A(f)$ | $\bar{A}$ |
| FedAvg | 36.68 | 59.17 | 68.12 | 80.34 | 29.02 | 53.89 | 63.06 | 78.27 |
| FL+EWC | 38.31 | 60.19 | 69.25 | 81.54 | 28.86 | 54.52 | 62.92 | 78.80 |
| Re-Fed | 38.08 | 59.02 | 67.85 | 79.85 | 28.82 | 52.84 | 61.53 | 76.69 |
| FOT | 40.18 | 61.41 | 68.54 | 79.70 | 31.31 | 55.56 | 62.41 | 76.32 |
| FedSSI | **42.58** | **62.65** | **72.09** | **82.49** | **32.95** | **56.12** | **64.91** | **79.10** |

## E.2  DETAILED RESULTS OF TEST ACCURACY

Table 8, 9, 10, 11 and 12 show the results of test accuracy on each incremental task in the **Acc** (Accuray) line. Here we measure average accuracy over all tasks on each client in the **Acc** line and highlight the best test accuracy in **bold**.

## E.3  DETAILED RESULTS OF COMMUNICATION ROUND.

Table 8, 9, 10, 11 and 12 show the detailed results of communication round on each incremental task in the **CoR** (Communication Round) line and highlight the results of the fewest number of communication rounds in underline.

Table 8: Performance comparisons of various methods on CIFAR10 with 5 incremental tasks.

| Method | Target | 2 | 4 | 6 | 8 | 10 | Avg |
|---|---|---|---|---|---|---|---|
| **CIFAR10** | | | | | | | |
| FedAvg | Acc | **88.43** | 70.23 | 55.00 | 45.53 | 36.68 | 59.17 |
| | CoR | 52 | 63 | 68 | 60 | 61 | 60.8 |
| FedProx | Acc | 88.43 | 69.62 | 56.50 | 45.58 | 35.88 | 59.20 |
| | CoR | 53 | 66 | 69 | 62 | 65 | 63 |
| FL+LwF | Acc | 88.43 | 70.98 | 56.10 | 46.12 | 38.04 | 59.93 |
| | CoR | 52 | 64 | 67 | 59 | 58 | 60.0 |
| FL+EWC | Acc | 88.43 | 71.32 | 56.60 | 46.30 | 38.31 | 60.19 |
| | CoR | 52 | 67 | 70 | 62 | 73 | 64.8 |
| FL+OGD | Acc | 88.43 | 71.10 | 56.20 | 46.14 | 37.55 | 59.88 |
| | CoR | 52 | 66 | 68 | 61 | 62 | 61.8 |
| FL+SI | Acc | 88.43 | 71.72 | 57.40 | 47.95 | 39.32 | 60.96 |
| | CoR | 52 | 64 | 67 | 59 | 54 | 59.2 |
| Re-Fed | Acc | 86.94 | 69.31 | 55.12 | 45.63 | 38.08 | 59.02 |
| | CoR | 59 | 67 | 68 | 60 | 65 | 63.80 |
| FedCIL | Acc | 85.73 | 68.44 | 54.60 | 44.77 | 37.96 | 58.30 |
| | CoR | 60 | 68 | 70 | 62 | 63 | 64.6 |
| GLFC | Acc | 87.16 | 70.42 | 57.27 | 46.89 | 38.43 | 60.03 |
| | CoR | 57 | 66 | 68 | 60 | 61 | 62.4 |
| FOT | Acc | 87.53 | 72.70 | 58.35 | 48.28 | 40.18 | 61.41 |
| | CoR | 55 | 65 | 68 | 60 | 58 | 61.2 |
| FedWeIT | Acc | 87.92 | 71.15 | 56.25 | 46.15 | 37.96 | 59.89 |
| | CoR | 56 | 66 | 67 | 61 | 62 | 62.4 |
| FedSSI | Acc | 88.29 | **72.58** | **59.43** | **50.35** | **42.58** | **62.65** |
| | CoR | 54 | 65 | 69 | 59 | 57 | 60.8 |

Table 9: Performance comparisons of various methods on CIFAR100 with 10 incremental tasks.

| Method | Target | 10 | 20 | 30 | 40 | 50 | 60 | 70 | 80 | 90 | 100 | Avg |
|---|---|---|---|---|---|---|---|---|---|---|---|---|
| **CIFAR100** | | | | | | | | | | | | |
| FedAvg | Acc | 69.12 | 55.48 | 49.32 | 42.75 | 38.56 | 35.21 | 33.10 | 31.23 | 29.67 | 27.15 | 41.36 |
| | CoR | 76 | 79 | 82 | 84 | 85 | 86 | 87 | 88 | 89 | 83 | 83.9 |
| FedProx | Acc | 68.62 | 54.90 | 48.95 | 42.45 | 38.34 | 35.10 | 33.00 | 31.12 | 28.88 | 27.84 | 40.92 |
| | CoR | 81 | 80 | 83 | 85 | 86 | 87 | 88 | 89 | 90 | 83 | 85.2 |
| FL+LwF | Acc | 69.12 | 56.01 | 50.32 | 44.75 | 39.89 | 36.54 | 34.12 | 32.34 | 30.67 | 31.91 | 42.56 |
| | CoR | 76 | 78 | 80 | 82 | 83 | 84 | 85 | 86 | 87 | 87 | 83.2 |
| FL+EWC | Acc | 69.12 | 56.34 | 50.98 | 45.21 | 40.45 | 37.01 | 34.58 | 32.98 | 32.47 | 33.36 | 43.25 |
| | CoR | 76 | 78 | 81 | 83 | 84 | 85 | 86 | 87 | 88 | 77 | 81 |
| FL+OGD | Acc | 69.12 | 56.45 | 51.12 | 45.78 | 41.12 | 37.45 | 35.00 | 34.57 | 32.13 | 32.87 | 43.56 |
| | CoR | 76 | 77 | 80 | 82 | 83 | 84 | 85 | 86 | 87 | 89 | 83.3 |
| FL+SI | Acc | 69.12 | 56.78 | 51.67 | 46.12 | 41.54 | 38.12 | 35.67 | 33.29 | 32.17 | 33.72 | 43.82 |
| | CoR | 76 | 78 | 80 | 82 | 83 | 84 | 85 | 86 | 87 | 75 | 81.6 |
| Re-Fed | Acc | 69.27 | 55.31 | 50.61 | 43.22 | 39.54 | 36.78 | 33.50 | 32.19 | 31.64 | 32.95 | 42.50 |
| | CoR | 82 | 80 | 82 | 84 | 83 | 86 | 85 | 87 | 87 | 88 | 84.4 |
| FedCIL | Acc | 68.49 | 56.06 | 49.76 | 43.98 | 38.82 | 36.93 | 33.28 | 32.30 | 31.07 | 30.88 | 42.16 |
| | CoR | 85 | 83 | 85 | 85 | 85 | 87 | 88 | 89 | 91 | 88 | 86.6 |
| GLFC | Acc | 68.98 | 56.67 | 50.22 | 44.09 | 39.21 | 37.43 | 35.16 | 34.53 | 33.34 | 33.17 | 43.28 |
| | CoR | 79 | 78 | 80 | 84 | 83 | 84 | 85 | 86 | 85 | 86 | 83 |
| FOT | Acc | 68.56 | 55.67 | 50.78 | 45.56 | 40.98 | 36.20 | 34.40 | 31.98 | 31.12 | 36.15 | 43.14 |
| | CoR | 79 | 78 | 81 | 82 | 84 | 85 | 86 | 87 | 88 | 90 | 84 |
| FedWeIT | Acc | **69.46** | 57.01 | 50.45 | 45.45 | 41.12 | 37.56 | 36.45 | 35.06 | 33.62 | 35.84 | 44.20 |
| | CoR | 88 | 86 | 87 | 88 | 89 | 90 | 90 | 90 | 90 | 83 | 87 |
| FedSSI | Acc | 69.12 | **57.34** | **52.12** | **47.01** | **43.12** | **40.01** | **37.45** | **35.23** | **33.45** | **37.96** | **45.28** |
| | CoR | 76 | 75 | 78 | 80 | 81 | 82 | 83 | 84 | 85 | 74 | 79.8 |

Table 10: Performance comparisons of various methods on Tiny-ImageNet with 10 incremental tasks.

| Method | Target | 20 | 40 | 60 | 80 | 100 | 120 | 140 | 160 | 180 | 200 | Avg |
|---|---|---|---|---|---|---|---|---|---|---|---|---|
| | | | | | | **Tiny-ImageNet** | | | | | | |
| FedAvg | Acc | 78.65 | 64.23 | 65.86 | 55.11 | 48.78 | 45.22 | 43.75 | 38.93 | 35.81 | 30.16 | 50.65 |
| | CoR | 95 | 87 | 92 | 84 | 91 | 89 | 86 | 88 | 94 | 87 | 89.3 |
| FedProx | Acc | 78.15 | 63.56 | 65.12 | 54.78 | 47.89 | 44.67 | 42.98 | 38.11 | 34.97 | 29.04 | 49.93 |
| | CoR | 97 | 88 | 93 | 85 | 90 | 88 | 87 | 89 | 93 | 90 | 90.0 |
| FL+LwF | Acc | 78.65 | 65.12 | 67.45 | 56.84 | 50.78 | 48.12 | 47.23 | 41.89 | 38.45 | 34.58 | 52.91 |
| | CoR | 95 | 89 | 94 | 86 | 92 | 90 | 89 | 90 | 91 | 81 | 89.7 |
| FL+EWC | Acc | 78.65 | 66.15 | 68.78 | 57.12 | 51.64 | 49.12 | 47.85 | 43.12 | 40.12 | 36.15 | 53.87 |
| | CoR | 95 | 88 | 93 | 87 | 91 | 90 | 88 | 91 | 92 | 94 | 90.9 |
| FL+OGD | Acc | 78.65 | 65.78 | 67.92 | 56.89 | 50.12 | 48.78 | 46.67 | 41.45 | 39.89 | 35.71 | 53.19 |
| | CoR | 95 | 89 | 94 | 86 | 92 | 90 | 89 | 90 | 91 | 84 | 90.0 |
| FL+SI | Acc | 78.65 | 66.12 | 68.45 | 57.34 | 51.12 | 49.18 | 47.89 | 41.82 | 40.04 | 35.87 | 53.65 |
| | CoR | 95 | 89 | 94 | 87 | 92 | 90 | 88 | 89 | 90 | 83 | 89.7 |
| Re-Fed | Acc | 78.54 | 65.27 | 66.39 | 55.21 | 49.36 | 47.45 | 45.83 | 40.32 | 38.03 | 33.43 | 51.98 |
| | CoR | 96 | 88 | 93 | 87 | 92 | 88 | 86 | 88 | 91 | 85 | 89.4 |
| FedCIL | Acc | **78.91** | 64.87 | 65.79 | 54.91 | 48.83 | 45.75 | 44.66 | 38.42 | 35.79 | 31.35 | 50.93 |
| | CoR | 96 | 89 | 92 | 86 | 91 | 90 | 89 | 90 | 91 | 89 | 90.3 |
| GLFC | Acc | 78.66 | 65.55 | 66.94 | 55.07 | 50.22 | 46.78 | 45.32 | 39.46 | 37.75 | 32.11 | 51.79 |
| | CoR | 96 | 88 | 94 | 87 | 91 | 89 | 87 | 89 | 90 | 90 | 90.1 |
| FOT | Acc | 78.84 | 66.45 | 69.08 | 58.89 | 53.07 | 51.06 | 48.15 | 43.78 | 42.32 | 37.23 | 54.87 |
| | CoR | 96 | 88 | 93 | 86 | 91 | 89 | 88 | 90 | 92 | 91 | 90.3 |
| FedWeIT | Acc | 78.51 | 65.78 | 67.85 | 57.34 | 48.78 | 48.34 | 47.45 | 41.45 | 39.78 | 34.98 | 53.04 |
| | CoR | 96 | 88 | 93 | 87 | 92 | 85 | 87 | 90 | 91 | 72 | 88.0 |
| FedSSI | Acc | 78.65 | **68.08** | **71.12** | **60.34** | **53.78** | **54.34** | **50.89** | **45.45** | **44.78** | **40.56** | **56.80** |
| | CoR | 95 | 89 | 95 | 87 | 92 | 90 | 88 | 91 | 92 | 87 | 90.6 |

Table 11: Performance comparisons of various methods on Digit10 with 4 domains.

| Method | Target | MNIST | EMNIST | USPS | SVHN | Avg |
|---|---|---|---|---|---|---|
| | | | **Digit10** | | | |
| FedAvg | Acc | **94.17** | 81.93 | 77.13 | 68.12 | 80.34 |
| | CoR | 58 | 52.6 | 49.9 | 47 | 51.9 |
| FedProx | Acc | 94.03 | 81.34 | 76.72 | 68.95 | 80.26 |
| | CoR | 56 | 51.6 | 49.5 | 47 | 51.0 |
| FL+LwF | Acc | 94.17 | 81.58 | 76.35 | 67.99 | 80.02 |
| | CoR | 58 | 53.2 | 51.1 | 49 | 52.8 |
| FL+EWC | Acc | 94.17 | 82.73 | 80.02 | 69.25 | 81.54 |
| | CoR | 58 | 60.4 | 58.0 | 56 | 58.1 |
| FL+OGD | Acc | 94.17 | 80.40 | 77.17 | 68.07 | 79.95 |
| | CoR | 58 | 53.4 | 51.5 | 50 | 53.2 |
| FL+SI | Acc | 94.17 | 81.86 | 77.84 | 69.79 | 80.92 |
| | CoR | 58 | 53.5 | 51.4 | 49 | 53.0 |
| Re-Fed | Acc | 93.47 | 80.92 | 77.15 | 67.85 | 79.85 |
| | CoR | 59 | 54.9 | 53.8 | 54 | 55.4 |
| FedCIL | Acc | 93.66 | 81.31 | 76.93 | 68.17 | 80.02 |
| | CoR | 56 | 56.3 | 54.7 | 53 | 55.0 |
| GLFC | Acc | 92.84 | 79.05 | 74.83 | 67.39 | 78.53 |
| | CoR | 57 | 56.4 | 54.2 | 52 | 54.9 |
| FOT | Acc | 93.04 | 80.25 | 76.98 | 68.54 | 79.70 |
| | CoR | 60 | 55.9 | 53.8 | 51 | 55.2 |
| FedWeIT | Acc | 93.97 | 82.15 | 77.81 | 69.71 | 80.91 |
| | CoR | 54 | 55.6 | 53.9 | 53 | 54.1 |
| FedSSI | Acc | 94.17 | **83.90** | **79.79** | **72.09** | **82.49** |
| | CoR | 58 | 57.9 | 55.6 | 53 | 56.1 |

Table 12: Performance comparisons of various methods on Office-31 with 3 domains and Office-Caltech-10 with 4 domains.

| Method | Target | Office31 | | | | Office-Caltech-10 | | | | |
|---|---|---|---|---|---|---|---|---|---|---|
| | | Amazon | Dlsr | Webcam | Avg | Amazon | Caltech | Dlsr | Webcam | Avg |
| FedAvg | Acc | **65.46** | 54.45 | 48.97 | 56.29 | 69.75 | 57.23 | 48.07 | 55.41 | 57.61 |
| | CoR | 56 | 51 | 58 | 55.0 | 36 | 24 | 38 | 34 | 33.0 |
| FedProx | Acc | 63.15 | 52.60 | 46.33 | 54.03 | 68.04 | 56.78 | 45.70 | 53.90 | 56.10 |
| | CoR | 57 | 54 | 55 | 55.3 | 36 | 35 | 37 | 37 | 36.2 |
| FL+LwF | Acc | 65.46 | 55.45 | 50.70 | 57.2 | 69.75 | 60.45 | 51.69 | 57.11 | 59.75 |
| | CoR | 56 | 52 | 54 | 54.0 | 36 | 32 | 34 | 32 | 33.5 |
| FL+EWC | Acc | 65.46 | 56.02 | 52.24 | 57.91 | 69.75 | 60.78 | 51.01 | 58.69 | 60.06 |
| | CoR | 56 | 52 | 58 | 55.3 | 36 | 38 | 39 | 37 | 37.5 |
| FL+OGD | Acc | 65.46 | 56.99 | 51.86 | 58.1 | 69.75 | 60.12 | 52.90 | 58.01 | 60.2 |
| | CoR | 56 | 53 | 55 | 54.7 | 36 | 36 | 37 | 39 | 37.0 |
| FL+SI | Acc | 65.46 | 56.29 | 53.10 | 58.28 | 69.75 | 61.01 | 58.50 | 51.82 | 60.27 |
| | CoR | 56 | 53 | 56 | 55.0 | 36 | 38 | 36 | 39 | 37.2 |
| Re-Fed | Acc | 65.26 | 57.02 | 50.11 | 57.46 | **69.82** | 59.74 | 51.30 | 59.16 | 60.01 |
| | CoR | 56 | 53 | 59 | 56 | 36 | 36 | 37 | 36 | 36.25 |
| FedCIL | Acc | 64.85 | 56.33 | 49.15 | 56.78 | 69.08 | 58.83 | 50.82 | 57.80 | 59.13 |
| | CoR | 55 | 56 | 57 | 56 | 35 | 37 | 38 | 37 | 36.75 |
| GLFC | Acc | 64.34 | 54.82 | 48.30 | 55.82 | 69.37 | 60.31 | 51.14 | 58.24 | 59.77 |
| | CoR | 56 | 55 | 55 | 55.3 | 35 | 38 | 39 | 36 | 37 |
| FOT | Acc | 64.76 | 54.62 | 49.12 | 56.17 | 69.45 | 61.45 | 51.82 | 60.30 | 60.76 |
| | CoR | 57 | 55 | 58 | 56.7 | 36 | 37 | 39 | 36 | 37.0 |
| FedWeIT | Acc | 64.98 | 57.02 | 51.49 | 57.83 | 69.14 | 60.22 | 50.99 | 58.53 | 59.72 |
| | CoR | 53 | 54 | 58 | 55.0 | 34 | 36 | 37 | 34 | 35.2 |
| FedSSI | Acc | 65.46 | **59.40** | **55.28** | **60.05** | 69.75 | **64.34** | **55.12** | **62.57** | **62.94** |
| | CoR | 56 | 60 | 58 | 58.0 | 36 | 35 | 38 | 34 | 35.8 |

