# OpenReview forum: "Rehearsal-Free Continual Federated Learning with Synergistic Regularization"
_ICLR.cc/2025/Conference — Submitted to ICLR 2025_

### Official Review · Reviewer_7XF1 · 2024-10-30

**Soundness:** 2
**Presentation:** 3
**Contribution:** 2
**Rating:** 6
**Confidence:** 4

**Summary:**

The paper addresses the challenge of Continual Federated Learning (CFL), which involves distributed devices learning from streaming tasks while retaining knowledge from previous tasks. The authors identify the limitations of existing CFL approaches that rely on data rehearsal and propose an regularization method called FedSSI, which enhances Synaptic Intelligence (SI) by incorporating a Personalized Surrogate Model (PSM). This approach allows clients to leverage both local and global data distributions, improving performance in heterogeneous data environments. The results demonstrate that FedSSI outperforms state-of-the-art methods.

**Strengths:**

1.The paper tackles the significant problem of catastrophic forgetting in Continual Federated Learning (CFL), which is crucial for the practical deployment of federated learning systems in dynamic environments.

2.The paper includes comprehensive experiments on various datasets and CFL task scenarios (both Class-Incremental Learning and Domain-Incremental Learning), providing strong empirical evidence of the method's effectiveness.

3. The use of personalized alternative models (PSM) to enhance synaptic intelligence (SI) can mitigate knowledge forgetting, especially in non-IID data Settings.

**Weaknesses:**

1.	While the paper tells an interesting story in this paper, the core idea is an extension of existing methods . It seems like a combination of Synaptic Intelligence (SI) and Continual Federated Learning (CFL).

2.	Although the authors' goal is to solve the CFL problem, it is actually the well-studied Non-iid problem in Federated Learning. This is also evident from the authors' theoretical analysis. There is no way for the authors to discuss how PSM is different under CFL.

3. The proposed method heavily depends on SI. If SI has inherent limitations or does not perform well under certain conditions, FedSSI might inherit these shortcomings.

4. The combination of global and local models is a well-explored area in federated learning, and the authors do not discuss a model mixing approach better suited to the characteristics of CFL, except for PSM. Proposing a new mixing method would help to increase the contribution of the paper.

5. Although the proposed method avoids data rehearsal, introducing the PSM could add storage overhead or complexity that may not be practical for edge devices with limited resources. The paper may not sufficiently discuss the storage costs or provide strategies to mitigate them.

**Questions:**

See weaknesses.

---

> ### Author Response · Authors · 2024-11-21
>
> Thanks for helpful comments from the esteemed reviewer! As for the revised manuscript, we will **highlight them in purple or blue** font in both the main text and the appendix for your convenience.
>
> > **W1&W2.  The novelty and detailed analysis of FedSSI. (blue in Line 300 and 337)**
>
> **R1:** Thank you for this valuable comment. **The unique challenges** in this paper are **more  comprehensive yet practical** than existing CFL works. In CFL, there are only a handful of works that successfully tackle both catastrophic forgetting and data heterogeneity simultaneously. Moreover, these existing works often rely on substantial resource expenditures and risk privacy breaches, employing techniques such as generative replay, memory buffers, and dynamic networks—primarily traditional centralized methods that overlook the resource constraints of client devices.
>
> **We believe** that a good paper often starts from practical applications to explore the underlying issues. In our article, we have explored the contradiction between the resource constraints of clients and the generally higher resource costs in continuous learning through empirical experiments. Our paper takes a different approach, **starting by** considering the **resource limitations** of client devices in FL and leveraging appropriate technologies. From this foundation, we aim to solve both catastrophic forgetting and data heterogeneity challenges concurrently. We hope our work will propel CFL research towards more practical and real-world scenarios.
>
> **Detailed Analysis:** Unlike SI algorithms in centralized environments that only consider the data distribution of a single environment, FedSSI introduces a Personalized Surrogate Model (PSM) to balance data heterogeneity across clients. Here, the PSM is **not used** as the target classification model; it is **solely** employed to calculate the contribution of parameters. Before clients receive new tasks, a PSM will be trained along with the global model on the current local task. Since this is purely local training assisted by an already converged global model, the training of the PSM is very fast (accounting for only **1/40** of the training cost per task and requiring no communication). We calculate and save the parameter contributions during the local convergence process of the PSM, which can then be locally **discarded** after its contribution has been computed. Then, each client trains on the new task with the local model and paprameter contribution scores.
>
> **At the technical level**, FedSSI's PSM **is not a simple combination** of Personalized Federated Learning (PFL) and Continual Learning (CL) techniques. PFL aims to train a personalized model for each client, which is the target classification model. Here, PSM serves as a computational auxiliary model for parameter contribution, skillfully balancing data heterogeneity while functioning as the model for calculating parameter contributions in SI. The use of this auxiliary model differs from the goals of PFL, exhibiting unique characteristics of CFL.
>
>
>
> > **W3. Concerns about the dependency of FedSSI on SI.**
>
> **R3:** Thank you for raising this concern.  Although the FedSSI algorithm uses PSM to improve the SI algorithm, the selection of the SI algorithm itself is not random. In our empirical experiments, we analyzed a vast number of existing CFL methods and techniques, as well as traditional CL techniques. The SI algorithm was chosen as the appropriate one among them, and it has been widely recognized for its efficiency and feasibility in most scenarios. We believe that this assumption is **acceptable**, similar to how many FL studies are based on CNN and ResNet series networks without considering the feasibility of techniques on a single Linear network.
>
>
>
> > **W4. Exploration of new model mixing methods for CFL.**
>
> **R4:** Thank you for your helpful comment. Similar to the response in **R1**, if it is merely a simple model mixing method, there have already been numerous studies on this in PFL, rendering it of little research value and difficult to achieve significant innovation. However, **at the technical level**, FedSSI's PSM **is not a simple combination** of Personalized Federated Learning (PFL) and Continual Learning (CL) techniques. PFL aims to train a personalized model for each client, which is the target classification model. Here, PSM serves as a computational auxiliary model for parameter contribution, skillfully balancing data heterogeneity while functioning as the model for calculating parameter contributions in SI. The use of this auxiliary model differs from the goals of PFL, exhibiting unique characteristics of CFL and better suited to the characteristics of CFL.

---

> > ### Author Response · Authors · 2024-11-21
> >
> > > **W5. Concerns about the addition storage cost by PSM. (purple in Line861-869)**
> >
> > **R5:** Thank you for this constructive suggestion.  As mentioned in **R1**, a PSM will be trained along with the global model on the current local task. Since this is purely local training assisted by an already converged global model, the training of the PSM is very fast (accounting for only **1/40** of the training cost per task and requiring no communication). We calculate and save the parameter contributions during the local convergence process of the PSM, which can then be locally **discarded** after its contribution has been computed. Then, each client trains on the new task with the local model and paprameter contribution scores.
> >
> > The model for an FL task is practically not large as edge clients are mostly resource-limited, and the memory demand by PSM is relatively similar to existing methods like FedProx as to manipulate over an extra model that has the same size as the federated model.
> >
> > **Possible Strategy:** If an FL task has large models and limited memory left on the edge, a prerequisite can be reasonably assumed satisfied: transmission capacity (probably after optimization) is sufficient for the system. In such a case, we can record PSM model in the server, which is only downloaded and updated in local to compute parameter contributions and is then uploaded to the server again.

---

> ### Comment · Reviewer_7XF1 · 2024-11-22
> **Comment**
>
> Thank you for your clear response. You have addressed all my concerns. Therefore, I decide to increase my score.

---

> > ### Author Response · Authors · 2024-11-22
> >
> > Sincere thanks for your response! We will further improve our manuscript in the final version.

---

### Official Review · Reviewer_EBSD · 2024-11-01

**Soundness:** 2
**Presentation:** 3
**Contribution:** 2
**Rating:** 6
**Confidence:** 4

**Summary:**

This work addresses the poor performance of Synaptic Intelligence in Non-IID scenarios within FL + CL contexts, where catastrophic forgetting remains significant. It attributes this to the failure of traditional global model-based surrogate loss methods due to data heterogeneity. Therefore, it introduces a personalized surrogate model to balance global and local knowledge during training.

**Strengths:**

1. The empirical results seem promising.
2. The proposed method is quite simple and the overall writing is clear.
3. The experimental analysis is abundant.

**Weaknesses:**

1. Some misunderstandings: A personalized surrogate model is introduced to balance global and local knowledge during training. However, it still requires data from previous tasks to update the personalized surrogate model, which seems to conflict with the "rehearsal-free" claim in the title. Especially in equation 5, the model needs the samples in last task to do the momentum updating in the new task. But I'm open to further discussions or authors' defending in their rebuttals.
2. Confusing definitions: I think it’s not accurate to conclude a class-incremental will share the same domain space. We can hardly say data from different classes will share a similar distribution. Despite that I do understand authors want to convey that data in class incremental setting would be data from one task, I think rewrite the definition part in 172-179 lines will be a wise choice.
3. Lack of in-depth analysis: The proposed method is more like a combination of existing techniques (personalized federated learning and Synaptic Intelligence). Plus, the analysis provided is not completely new to me as Synaptic Intelligence is a very matured technique in Continual Learning. From my perspective, the fact that data heterogeneity will harm the effects of regulation-based anti-forgetting techniques is also quite normal. I do not find adding a personalized surrogate model for each client and using Synaptic Intelligence technique is a challenge. Of course, I acknowledge that employing PSM module based on SI algorithm to further alleviate the catastrophic forgetting caused by data heterogeneity is somewhat new but it may not reach an ICLR standard.

If authors are willing to defend their points (especially on 3) and provide more insights on what's the key challenges in CFL and combining PSM module based on SI algorithm, I willing to raise my scores based on the further discussions.

**Questions:**

See the weakness.

---

> ### Author Response · Authors · 2024-11-21
>
> Thanks for helpful comments from the esteemed reviewer! As for the revised manuscript, we will **highlight them in green or blue** font in both the main text and the appendix for your convenience.
>
> > **W1. Concerns about the rehearsal-free claim. (green in Line 309)**
>
> **R1:** Thank you for raising this concern. I apologize for the ambiguity in our expression that caused your misunderstanding. In FedSSI, the PSM is **not used** as the target classification model; it is **solely** employed to calculate the contribution of parameters. Before clients receive new tasks, a PSM will be trained along with the global model on the current local task. Since this is purely local training assisted by an already converged global model, the training of the PSM is very fast (accounting for only **1/40** of the training cost per task and requiring no communication). We calculate and save the parameter contributions during the local convergence process of the PSM, which can then be locally **discarded** after its contribution has been computed. Then, each client trains on the new task with the local model and paprameter contribution scores.
>
> In Equation.5, we used "previous task" to distinguish it from a "new task," but in reality, the "new task" had **not yet arrived**, and the "previous task" in the original text actually referred to the **current task**. We have revised the ambiguous text.
>
>
>
> > **W2. Confusing definitions about the class-incremental learning. (green in Line 172-175)**
>
> **R2:** Thank you for this constructive comment. We are pleasantly surprised to receive such constructive suggestions, which is unprecedented. We have revised the original description and decided to **abandon** the statement that class-incremental tasks completely share the same domain. Instead, we have modified it to state that the **main challenge** in this scenario stems from **changes in the number of classes**, which we believe is a more precise formulation (we hope to obtain your confirmation).
>
>
>
> > **W3. Lack of in-depth analysis about FedSSI.(blue in Line 300 and 337)**
>
> **R3:** Thank you for raising this concern. **The unique challenges** in this paper are **more  comprehensive yet practical** than existing CFL works. In CFL, there are only a handful of works that successfully tackle both catastrophic forgetting and data heterogeneity simultaneously. Moreover, these existing works often rely on substantial resource expenditures and risk privacy breaches, employing techniques such as generative replay, memory buffers, and dynamic networks—primarily traditional centralized methods that overlook the resource constraints of client devices.
>
> **We believe** that a good paper often starts from practical applications to explore the underlying issues. In our article, we have explored the contradiction between the resource constraints of clients and the generally higher resource costs in continuous learning through empirical experiments. Our paper takes a different approach, **starting by** considering the **resource limitations** of client devices in FL and leveraging appropriate technologies. From this foundation, we aim to solve both catastrophic forgetting and data heterogeneity challenges concurrently. We hope our work will propel CFL research towards more practical and real-world scenarios.
>
> **At the technical level**, FedSSI's PSM **is not a simple combination** of Personalized Federated Learning (PFL) and Continual Learning (CL) techniques. PFL aims to train a personalized model for each client, which is the target classification model. Here, PSM serves as a computational auxiliary model for parameter contribution, skillfully balancing data heterogeneity while functioning as the model for calculating parameter contributions in SI. The use of this auxiliary model differs from the goals of PFL, exhibiting unique characteristics of CFL.

---

> > ### Comment · Reviewer_EBSD · 2024-11-22
> >
> > Thank you for the authors' rebuttal. They have addressed my concerns regarding the rehearsal-free claim and the unclear definitions of class-incremental learning. I also reviewed the revised manuscript, and I believe it has improved the clarity and readability of the paper. Some of concerns on the core novelty still remains, after reading other reviewer's comments I find they also raise the concern on this point. But I really appreciate author's efforts, now I think "PSM serves as a computational auxiliary model for parameter contribution, skillfully balancing data heterogeneity while functioning as the model for calculating parameter contributions in SI" do have some new insights. Thus, I will consider to raise my rate after discussing with other reviewers.

---

> > > ### Author Response · Authors · 2024-11-22
> > >
> > > Sincere thanks for your response! We deeply appreciate your conscientiousness and responsibility in the review process, and we agree with your viewpoints. Currently, except for Reviewer y9b8, all other reviewers have decided to raise the score from 5 to 6. After reading their comments, we understand that they still think that our paper may require more novelty. Nevertheless, they also acknowledge our work, believing that it meets the standards of ICLR. We hope you can reconsider our score based on the reviewers' responses. Although the esteemed reviewer tends to increase the score after discussion, most reviewers have made their judgments and decided to raise our score. Thank you very much for your valuable suggestions and professional review attitude!

---

> > > ### Author Response · Authors · 2024-11-28
> > >
> > > Sorry for disturbing the esteemed reviewer!  As the discussion phase draws close, except for Reviewer y9b8, all other reviewers have raised the score from 5 to 6. After reading their comments, we understand they still think our paper may require more novelty. Nevertheless, they also acknowledge our work, believing that it meets the standards of ICLR. We hope you can reconsider our score based on the reviewers' responses. Thank you for your valuable suggestions and professional review.

---

> > > > ### Comment · Reviewer_EBSD · 2024-12-03
> > > >
> > > > I note that there has been no further discussion among reviewers to address the previously raised concerns. While the paper shows some improvement, several issues mentioned in the earlier review stage remain unresolved. But I do appreciate authors' efforts to improve this paper. Given the current state of the manuscript, I am willing to adjust my score to 6. However, addressing the remaining concerns would further strengthen the paper.

---

> > > > > ### Author Response · Authors · 2024-12-03
> > > > >
> > > > > Sincere thanks for your response! We will further improve our manuscript in the final version.

---

### Official Review · Reviewer_ZjRU · 2024-11-03

**Soundness:** 3
**Presentation:** 4
**Contribution:** 2
**Rating:** 6
**Confidence:** 3

**Summary:**

The paper introduces FedSSI, a method for tackling Continual Federated Learning (CFL) in federated environments. In CFL, clients continuously learn from new data while retaining knowledge from previous tasks, all without directly sharing data to ensure privacy. A common solution to this problem involves data rehearsal, where past samples are stored and replayed to maintain learning continuity. However, this method can be impractical due to memory limitations. FedSSI addresses these issues by enhancing synaptic intelligence, a technique originally developed for continual learning, to work in CFL settings without the need for data rehearsal. FedSSI has  Personalized Surrogate Model (PSM), which allows each client to balance local data with global information during training. This dual knowledge approach helps clients to mitigate forgetting of previous tasks and enables efficient learning even with highly diverse data across clients. Through extensive experimentation, FedSSI demonstrates its effectiveness over existing methods, particularly in scenarios with significant data heterogeneity, where data differences between clients can be a major obstacle to model performance. By forgoing rehearsal and reducing memory overhead, FedSSI offers a more practical and resource-efficient solution to CFL, aligning well with the demands of real-world applications where data privacy, device limitations, and diverse client data must be managed concurrently.

**Strengths:**

1. FedSSI eliminates the need for data rehearsal, addressing reducing memory usage on client devices. This makes it highly suitable for resource-constrained environments in FL.

2. The proposed method effectively handles data heterogeneity across clients by introducing the personalized surrogate model (PSM). The model remains accurate and robust even when clients have significantly diverse data (non-IID).

3. Authors have included extensive experiments across diverse datasets and CFL task scenarios.

**Weaknesses:**

1. The core novelty is incremental rather than foundational. The work builds on known methods (SI) by adjusting them for CFL with a new model component (Personalized Surrogate Model). This makes the paper valuable but not groundbreaking. What are the unique challenges the author faced while applying SI to CFL (Apart from data heterogeneity (non-IID) and catastrophic forgetting)?

2. Can the author provide the scalability of the proposed method? What is the total number of clients on whom the method is tested? What if clients participate only partially, i.e., clients are not available for all global iterations? Authors can perform experiments on multiple clients, say 100, and 10% of the clients are available per iteration.  From here, it would be clear the proposed approach is applicable in cross-silo as well as cross-device FL scenarios.

3. In Table 3, what is the relation between $\alpha$ and $\lambda$? Can the authors explain it? Is there any theoretical relation between these two?

4. In Table 4, the authors evaluate the best test accuracy versus the communication rounds. Except for CIFAR100,  the proposed approach takes more communication rounds to achieve the best accuracy. Can the author explain why the proposed approach needs more communication rounds? Authors can provide a more detailed analysis of the trade-offs between communication rounds and accuracy. Authors can give an insight into the impact of the method's practical implementation in bandwidth-constrained environments.

**Questions:**

Asked in weakness.

---

> ### Author Response · Authors · 2024-11-21
>
> Thanks for helpful comments from the esteemed reviewer! As for the revised manuscript, we will **highlight them in red or blue** font in both the main text and the appendix for your convenience.
>
> > **W1. Concerns about novelty and unique challenges. (blue in Line 300 and 337)**
>
> **R1:** Thank you so much for raising this concern. **Novelty:** This paper originates from an analysis of existing Continual Federated Learning (CFL) methods, focusing on the issue of resource constraints at the client side. **Such empirical analysis is forward-looking and significant within the field of CFL, reflecting the advanced nature of our motivation.** However, under the condition of considering resource constraints, CFL methods fail to effectively address both catastrophic forgetting and data heterogeneity, which are core issues in continual learning and federated learning respectively. FedSSI first empirically explores and demonstrates the superiority of the Synaptic Intelligence (SI) algorithm in mitigating catastrophic forgetting and reducing resource consumption.
>
> Building on this, it employs a Parameter Sharing Mechanism (PSM) to further address the issue of data heterogeneity. **At the technical level**, FedSSI's PSM **is not a simple combination** of Personalized Federated Learning (PFL) and Continual Learning (CL) techniques. PFL aims to train a personalized model for each client, which is the target classification model. Here, PSM serves as a computational auxiliary model for parameter contribution, skillfully balancing data heterogeneity while functioning as the model for calculating parameter contributions in SI. The use of this auxiliary model differs from the goals of PFL, exhibiting unique characteristics of CFL.
>
> **The unique challenges** in this paper are **more  comprehensive yet practical** than existing CFL works. In CFL, there are only a handful of works that successfully tackle both catastrophic forgetting and data heterogeneity simultaneously. Moreover, these existing works often rely on substantial resource expenditures and risk privacy breaches, employing techniques such as generative replay, memory buffers, and dynamic networks—primarily traditional centralized methods that overlook the resource constraints of client devices. **We believe** that a good paper often starts from practical applications to explore the underlying issues. In our article, we have explored the contradiction between the resource constraints of clients and the generally higher resource costs in continuous learning through empirical experiments. Our paper takes a different approach, **starting by** considering the **resource limitations** of clients in FL and leveraging appropriate technologies. From this foundation, we aim to solve both catastrophic forgetting and data heterogeneity challenges concurrently. We hope our work will propel CFL research towards more practical and real-world scenarios.
>
>
>
> > **W2. Scalability of FedSSI and Configurations. (red in Line 873-895)**
>
> **R2:** Thanks a lot for your comment. At line 374, we described the experimental Configuration and noted that the remaining details are provided in Table 6 (Appendix). We set up different experimental environments based on the size of the datasets (for example, CIFAR10 used 20 clients with a client selection rate of 40%, i.e., 8 clients per round of communication), which is a common FL setting. In response to the scalability concern raised by the esteemed reviewer, we conducted further experiments by increasing the number of clients to 100 while reducing the client selection rate to 10%. We performed some related experiments on CIFAR10 and Digit10 ($\alpha=10.0$), with the results as follows:
>
> |             |  Metric   | FedAvg | FL+EWC | Re-Fed |  FOT  |  FedSSI   |
> | :---------: | :-------: | :----: | :----: | :----: | :---: | :-------: |
> | **CIFAR10** |  $A(f)$   | 18.67  | 19.93  | 19.44  | 21.26 | **23.61** |
> |             | $\bar A$  |  45.8  | 46.38  | 44.08  | 47.02 | **47.14** |
> | **Digit10** | $ A(f) $  | 55.91  | 56.82  | 54.91  | 56.06 | **59.35** |
> |             | $ \bar A$ | 70.37  |  70.4  | 66.24  | 69.69 | **71.27** |
>
> Since the dataset needs to be divided into different numbers of tasks, an excessive number of clients can lead to a very small number of samples per client, making model training difficult. However, **FedSSI still maintains a leading position.** To better address this issue, we will conduct subsequent experiments on larger datasets such as DomainNet to ensure the accuracy of cross-silo validation. (Due to the longer training time, we regret that we cannot provide experimental results on large-scale datasets during the rebuttal period.)

---

> > ### Author Response · Authors · 2024-11-21
> >
> > > **W3. The relation between α and λ in Table 3. (red in Line 513-516)**
> >
> > **R3:** Thanks a lot for raising this concern. *α* refers to the degree of data heterogeneity, while *λ* is a control coefficient in the training process of PSM. In Proposition 1, we show that adjusting *λ* can control whether the proportion of knowledge in PSM leans towards the local distribution or the global distribution (i.e., it is related to *α*). When *α* has a higher value, indicating a trend towards homogeneity in distribution, clients need to focus more on local knowledge. This means that by setting a larger *λ* value, PSM can rely more on local knowledge. Although we cannot directly relate *α* and *λ* with a simple formula due to the complexity of the problem, even in specialized research on personalized federated learning (PFL), methods such as Gaussian mixture modeling are relied upon. In this paper, we can **empirically and theoretically** judge there exists a positive correlation between *α* and *λ*, which is supported by Proposition 1 and extensive experiments conducted in Table 3.
> >
> >
> >
> > > **W4.  Explaination for communication efficency and Insight into the implementation with constrained bandwidth. (red in Line 474-482, 487-514 and 873-895)**
> >
> > **R4:** Thank you so much for pointing out this problem. In Table 4, we report the rounds required for each method to achieve the best test accuracy presented in Table 1. However, since the best test accuracy values differ among methods, a direct comparison of these rounds is inappropriate. Instead, we should compare the **difference between the accuracy improvement percentage and the round increase percentage of FedSSI and other baselines**. We observed that FedSSI can achieve a significant $\Delta$ value for all datasets except Digit10, indicating that our communication method is efficient and can bring considerable performance improvements. For Digit10, a possible reason is that the dataset itself has simple features (one-channel images), and all methods can achieve a relatively good accuracy. In such cases, the percentage increase in accuracy by FedSSI is relatively small. However, achieving a high performance improvement on a simple dataset is not easy in itself. In the next version, we will consider using a smaller network model (as learning Digit10 with a CNN is sufficient) for verification. **We have revised the description of Table 4 by adding the analysis of trades-off communication rounds and performance improvement to demonstrate the communication efficiency of each method.** Detailed description on the communication rounds for each task across different datasets for each method are provided in the Appendix.
> >
> > |             |  Metric   | FedAvg | FL+EWC | Re-Fed |  FOT  |  FedSSI   |
> > | :---------: | :-------: | :----: | :----: | :----: | :---: | :-------: |
> > | **CIFAR10** |  $A(f)$   | 29.02  | 28.86  | 28.82  | 31.31 | **32.95** |
> > |             | $\bar A$  | 53.89  | 54.52  | 52.84  | 55.56 | **56.12** |
> > | **Digit10** |  $A(f)$   | 63.06  | 62.92  | 61.53  | 62.41 | **64.91** |
> > |             | $ \bar A$ | 78.27  |  78.8  | 76.69  | 76.32 | **79.10** |
> >
> > To further explore the impact of bandwidth on FedSSI, we have already conducted experiments with an increased number of clients in **W2**. Here, we continue to follow the client number settings described in the paper, but we **halve both the client selection rate and the communication rounds** for experiments on CIFAR10 and Digit10. The experiments show that although the performance of all methods declines, FedSSI maintains its **leading position**.

---

> > > ### Comment · Reviewer_ZjRU · 2024-11-22
> > >
> > > Thank you for the authors' rebuttal. They have addressed my concerns in the reviews. The author tried to explain the novelty of the paper. I still feel the novelty is not foundational but relatively incremental.  However, I consider this paper a good one and thus increase my score.
> > >
> > > Best of luck!

---

> > > > ### Author Response · Authors · 2024-11-22
> > > >
> > > > Sincere thanks for your response! Although we cannot fundamentally alter our paper we will refine our manuscript to showcase its novelty further and strive for more groundbreaking work in our future works.

---

### Official Review · Reviewer_y9b8 · 2024-11-04

**Soundness:** 2
**Presentation:** 2
**Contribution:** 2
**Rating:** 5
**Confidence:** 3

**Summary:**

The paper proposes Federated Synaptic Synergistic Intelligence (FedSSI), a continual federated learning (CFL) approach aimed at reducing catastrophic forgetting without relying on memory-intensive rehearsal techniques. FedSSI employs a regularization-based mechanism named personalized surrogate model (PSM) integrated with Synaptic Intelligence (SI) to handle data heterogeneity across clients in a federated setting. The approach is evaluated across several benchmarks, where it reportedly outperforms baseline CFL methods.

**Strengths:**

1. The paper addresses a well-known issue in CFL: catastrophic forgetting in the absence of a centralized memory. The rehearsal-free approach aligns well with the privacy and memory constraints of federated learning.

2. The authors validate FedSSI through extensive experiments across multiple benchmarks and tasks, including comparisons with several state-of-the-art CFL methods. The results consistently demonstrate FedSSI’s effectiveness, highlighting its potential as a valuable addition to the CFL landscape.

**Weaknesses:**

1. Clarity of the proposed method needs to be improved. The proposed method mainly consists of a new way to compute the contribution of the $i$-th parameter in client $k$ to the change of the loss function used in SI and tailored for the CFL problem. I find it very hard to understand intuitively or theoretically why the proposed method is able to address data heterogeneity. Without a strong theoretical justification or comprehensive analysis, the approach lacks depth and could be viewed as heuristic. Moreover,

2. The proposed FedSSI approach builds on well-known regularization techniques commonly used in continual learning, i.e., Synaptic Intelligence. The claim of being rehearsal-free as a unique contribution is underwhelming without showing a clear, unique mechanism that fundamentally sets FedSSI apart from similar approaches. The rehearsal-free property of the proposed method comes from the use of SI as well. By combining elements of personalization and regularization, FedSSI does not seem to introduce fundamentally new ideas but rather adapts existing ones for CFL.

3. Minor: there are many reference issues in the paper.

**Questions:**

I suggest the authors to add an algorithm of the proposed method.

---

> ### Author Response · Authors · 2024-11-21
>
> Thanks for helpful comments from the esteemed reviewer! As for the revised manuscript, we will **highlight them in blue** font in both the main text and the appendix for your convenience.
>
> > **W1.  Hard to understand why FedSSI can address data heterogeneity. (blue in Line 300 and 322)**
>
> **R1:** Thanks a lot for raising this concern. FedSSI can **theoretically** be proven to address the data heterogeneity. Considering that our writing might lead to misunderstandings on the part of the esteemed reviewer, I will briefly reintroduce the core of our method here:
>
> Unlike SI algorithms in centralized environments that only consider the data distribution of a single environment, FedSSI introduces a Personalized Surrogate Model (PSM) to balance data heterogeneity across clients. Here, the PSM is **not used** as the target classification model; it is **solely** employed to calculate the contribution of parameters. Before clients receive new tasks, a PSM will be trained along with the global model on the current local task. Since this is purely local training assisted by an already converged global model, the training of the PSM is very fast (accounting for only **1/40** of the training cost per task and requiring no communication). We calculate and save the parameter contributions during the local convergence process of the PSM, which can then be locally **discarded** after its contribution has been computed. Then, each client trains on the new task with the local model and paprameter contribution scores. We have supplemented the algorithm in **Q1** for your better understanding.
>
> The training of the PSM is defined in Equation 5. Denote that *α* refers to the degree of data heterogeneity, while *λ* is a control coefficient in the training process of PSM. In Proposition 1, we show that adjusting *λ* can control whether the proportion of knowledge in PSM leans towards the local distribution or the global distribution (i.e., it is related to *α*). When *α* has a higher value, indicating a trend towards homogeneity in distribution, clients need to focus more on local knowledge. This means that by setting a larger *λ* value, PSM can rely more on local knowledge. Thus, we can **empirically and theoretically** judge there exists a positive correlation between *α* and *λ*, which is supported by Proposition 1 and extensive experiments conducted in Table 3.
>
>
>
> > **W2. Concerns about the novelty and unique mechanism of FedSSI. (blue in Line 337)**
>
> **R2:**  Thank you for raising such an insightful comment. This paper originates from an analysis of existing Continual Federated Learning (CFL) methods, focusing on the issue of resource constraints at the client side. **We believe** that a good paper often starts from practical applications to explore the underlying issues. In our article, we have explored the contradiction between the resource constraints of clients and the generally higher resource costs in CL through empirical experiments. **Such empirical analysis is forward-looking and significant within the field of CFL, reflecting the advanced nature of our motivation.**
>
> However, under the condition of considering resource constraints, CFL methods fail to effectively address both catastrophic forgetting and data heterogeneity, which are core issues in continual learning and federated learning respectively. FedSSI first empirically explores and demonstrates the superiority of the Synaptic Intelligence (SI) algorithm in mitigating catastrophic forgetting and reducing resource consumption. Building on this, it employs a Parameter Sharing Mechanism (PSM) to further address the issue of data heterogeneity.
>
> At the technical level, FedSSI's PSM **is not a simple combination** of Personalized Federated Learning (PFL) and Continual Learning (CL) techniques. PFL aims to train a personalized model for each client, which is the target classification model. Here, PSM serves as a computational auxiliary model for parameter contribution, skillfully balancing data heterogeneity while functioning as the model for calculating parameter contributions in SI. The use of this auxiliary model differs from the goals of PFL, exhibiting unique characteristics of CFL.
>
>
>
> > **W3 & Q1. Reference issues and algorithm. (blue in Line 32 and 270)**
>
> **R3:**  Thanks for this constructive suggestion. We have provided an algorithm to illustrate the procedure of FedSSI. Regarding the issue of references, we hope to receive guidance from the esteemed reviewer, but we  have revised a mistake in Introduction section and will thoroughly examine this matter later and make corrections in subsquent versions.

---

> ### Author Response · Authors · 2024-11-28
>
> Dear reviewer y9b8, during this rebuttal period, we have:
>
> - Explained how our proposed approach addresses the issue of data heterogeneity with both theoretical and empirical analysis.
> - Clarified the unique challenge and novelty of our method
> - Revised reference issues and provided the algorithm description.
>
> In this rebuttal, all the main concerns the esteemed reviewers raised are similar, and our responses have received acknowledgment and score improvements from all the remaining reviewers. We would love to hear your feedback on our updates and look forward to discussing any remaining concerns you may have. Thank you for your time and consideration.

---

> ### Author Response · Authors · 2024-12-03
>
> Dear Reviewer y9b8,
>
> I hope this comment finds you well. During the rebuttal and discussion phrase, **all the main concerns the esteemed reviewers raised are similar, and our responses have received acknowledgment and score improvements from all other three reviewers**. Since there are only a few hours left until the end of the discussion,we would love to hear your feedback on our updates and look forward to discussing any remaining concerns you may have. Thank you for your time and consideration.
>
> Authors,

---

### Author Response · Authors · 2024-11-21
**Response to All Reviewers**

We deeply appreciate the **valuable** suggestions from every reviewer, which are all **professional, sincere, and conducive** to enhancing the quality of our paper. We have responded **diligently** to **every question raised by each reviewer** and conducted supplementary experiments as well as revised the paper, hoping to further discuss with you. We apologize for any inadequacies in our description of the method and innovative points, which we have significantly adjusted in the revised manuscript and individually addressed in our responses to each esteemed reviewer.

---

> ### Author Response · Authors · 2024-11-24
>
> Thank you once again to all the reviewers for your diligent efforts during the review and discussion! As the discussion phase draws to a close, considering the new insights into the novelty of FedSSI and the improved presentation in our revised manuscript and rebuttal, we are summarizing here to facilitate the understanding by the esteemed reviewers:
>
> **Motivation:** In CFL, only a limited number of works have successfully addressed both catastrophic forgetting and data heterogeneity simultaneously. Furthermore, these existing works often rely on significant resource overheads and pose risks of privacy breaches, utilizing techniques such as generative replay, memory buffers, and dynamic networks—primarily traditional centralized methods that neglect the resource constraints of client devices. **Our paper** adopts a different approach, starting by acknowledging the resource limitations of client devices in FL and leveraging suitable technologies. Based on this foundation, we aim to tackle both catastrophic forgetting and data heterogeneity challenges concurrently. We hope our work will advance CFL research towards more practical and real-world applications.
>
> **Technique Improvement:** Our improvement is not trivial but rather simple and efficient. We have incorporated the PSM module with SI, which is not merely a straightforward combination of PFL and CL techniques. PFL aims to train a personalized model for each client, which serves as the target classification model. In this context, PSM functions as a computational auxiliary model for parameter contribution, skillfully balancing data heterogeneity while also acting as the model for calculating parameter contributions in SI process. The utilization of this auxiliary model diverges from the objectives of PFL, demonstrating the unique characteristics of CFL.

---

### Author Response · Authors · 2024-12-03

Dear Reviewers and Area Chair,

Thank you very much for taking the time to read our revised manuscript! We appreciate our continued engagement and the valuable feedback/comments during this period. We are very happy to see that **reviewers ZjRU, 7XF1 and EBSD** are satisfied with our responses and have raised their scores during this discussion phase. It is sad that **reviewer y9b8** has not yet responded during the rebuttal period. However, his questions are similar to those of other reviewers, and I believe that if he could see our rebuttal, he would also consider raising the score.

We are thankful to the reviewers for raising their scores. If the reviewers are satisfied with our updates and responses, and believe that our work is worthy of acceptance, we kindly ask them to consider rating our paper as a "full accept". We remain available to resolve any outstanding issues that would improve our paper from a borderline to an accept in the remaining time allotted for discussion.

Thank you again for your time and consideration,

Authors

---

### Meta-Review · Area_Chair_TmPt · 2024-12-19

**Metareview:**

This paper considers federated continual learning setting which combines the continual learning setting and federated learning setting. The proposed approach is an extension of an existing regularization-based approach called synaptic intelligence which assigns individual importance weights to each dimension of the parameters in the regularization. The paper propose a different strategy for computing the importance weights to account for the mis-match between local and global model using the so-called personalized surrogate model.

Although the paper is well-motivated, the reviewers and AC found the novelty of the paper is quite limited. In addition, the technical analysis of the proposed approach is trivial without giving any insights. The main proposition 1 just shows when the the regularization parameter is larger enough the localized model converge to the global model, which does not help people to understand the proposed personalized surrogate model is better for learning the importance weights. Given this critical weakness, I would recommend a rejection due to the high quality bar of ICLR.

**Additional Comments On Reviewer Discussion:**

The reviewers have engaged with discussion with authors and acknowledged the additional efforts made by the authors during the rebuttal period. However, the main concerns about limited novelty and lack of technical analysis remain.

---

### Decision · Program_Chairs · 2025-01-22

Reject